# Statistical Guarantees for High-Dimensional Stochastic Gradient Descent

**Jiaqi Li**
Department of Statistics
University of Chicago
Chicago, IL 60637
jqli@uchicago.edu

**Zhipeng Lou**
Department of Mathematics
University of California, San Diego
La Jolla, CA 92093
zlou@ucsd.edu

**Johannes Schmidt-Hieber**
Department of Applied Mathematics
University of Twente
Enschede, Netherlands
a.j.schmidt-hieber@utwente.nl

**Wei Biao Wu**
Department of Statistics
University of Chicago
Chicago, IL 60637
wbwu@uchicago.edu

## Abstract

Stochastic Gradient Descent (SGD) and its Ruppert–Polyak averaged variant (ASGD) lie at the heart of modern large-scale learning, yet their theoretical properties in high-dimensional settings are rarely understood. In this paper, we provide rigorous statistical guarantees for constant learning-rate SGD and ASGD in high-dimensional regimes. Our key innovation is to transfer powerful tools from high-dimensional time series to online learning. Specifically, by viewing SGD as a nonlinear autoregressive process and adapting existing coupling techniques, we prove the geometric-moment contraction of high-dimensional SGD for constant learning rates, thereby establishing asymptotic stationarity of the iterates. Building on this, we derive the $q$-th moment convergence of SGD and ASGD for any $q \geq 2$ in general $\ell^s$-norms, and, in particular, the $\ell^\infty$-norm that is frequently adopted in high-dimensional sparse or structured models. Furthermore, we provide sharp high-probability concentration analysis which entails the probabilistic bound of high-dimensional ASGD. Beyond closing a critical gap in SGD theory, our proposed framework offers a novel toolkit for analyzing a broad class of high-dimensional learning algorithms.

## 1   Introduction

Stochastic gradient descent (SGD) has been a cornerstone in large-scale machine learning since the seminal work by Robbins and Monro [1951]. It is especially efficient in high-dimensional and overparameterized settings where the number of unknown parameters can exceed the number of training samples [Arpit et al., 2017, Zhang et al., 2017, He et al., 2016]. SGD can also be combined with regularization techniques such as dropout to prevent overfitting in large networks [Krizhevsky et al., 2012, Srivastava et al., 2014]. Despite the vast amount of theoretical work on SGD, generalization bounds of SGD in high-dimensional regimes remain limited [Garrigos and Gower, 2023]. Considering a strongly convex objective function, we provide statistical guarantees for constant learning-rate SGD and its Ruppert–Polyak averaged variant (ASGD) [Ruppert, 1988, Polyak and Juditsky, 1992] in high-dimensional settings.

39th Conference on Neural Information Processing Systems (NeurIPS 2025).

Specifically, we consider a general optimization problem

$$\boldsymbol{\beta}^* \in \arg\min_{\boldsymbol{\beta} \in \mathbb{R}^d} G(\boldsymbol{\beta}), \text{ where } \boldsymbol{\beta} \mapsto G(\boldsymbol{\beta}) := \mathbb{E}_{\boldsymbol{\xi} \sim \Pi} g(\boldsymbol{\beta}, \boldsymbol{\xi}), \tag{1}$$

$g(\cdot)$ is the noise-perturbed measurement of $G(\cdot)$, and $\boldsymbol{\xi}$ denotes a random element sampled from some unknown distribution $\Pi$. Given i.i.d. random samples $\boldsymbol{\xi}_1, \boldsymbol{\xi}_2, \ldots$ and some initialization $\boldsymbol{\beta}_0 \in \mathbb{R}^d$, the $k$-th SGD iteration is

$$\boldsymbol{\beta}_k = \boldsymbol{\beta}_{k-1} - \alpha \nabla g(\boldsymbol{\beta}_{k-1}, \boldsymbol{\xi}_k), \quad k = 1, 2, \ldots, \tag{2}$$

for some constant learning rate $\alpha > 0$, and $\nabla g(\boldsymbol{\beta}, \boldsymbol{\xi}) = \nabla_{\boldsymbol{\beta}} g(\boldsymbol{\beta}, \boldsymbol{\xi})$ the stochastic gradient with respect to $\boldsymbol{\beta}$. For $k \geq 1$, the ASGD variant is defined by

$$\bar{\boldsymbol{\beta}}_k = \frac{1}{k} \sum_{i=1}^{k} \boldsymbol{\beta}_i. \tag{3}$$

We are interested in the high-dimensional setting where the parameter dimension $d$ can be very large. Here, a notable divide between empirical success and theoretical understanding is that practitioners often employ a large constant learning rate $\alpha$ in (2) to accelerate convergence in high-dimensional problems [Wu et al., 2018, Cohen et al., 2021, Cai et al., 2024]. However, such choices can induce pronounced non-stationarity in the SGD iterates $\{\boldsymbol{\beta}_k\}_{k\in\mathbb{N}}$ which will not converge to a point but oscillates around the mean of a stationary distribution. In other words, $\boldsymbol{\beta}_k$ is non-stationary but asymptotically stationary, which converges only in distribution as $k \to \infty$, while the mean of this distribution differs from the exact minimizer $\boldsymbol{\beta}^*$ due to the non-diminishing bias of order $O(\alpha)$ [Dieuleveut et al., 2020, Merad and Gaïffas, 2023]. Classical theory mostly relies on decaying learning rates [Zhang, 2004, Nemirovski et al., 2009, Jentzen and von Wurstemberger, 2020, Shi et al., 2023]. To address the non-stationarity issue, we apply powerful tools from nonlinear time series analysis [Wu and Shao, 2004] to online learning, particularly by adapting the coupling techniques to show the geometric-moment contraction of SGD for constant learning rates. Specifically, for any two SGD sequences $\{\boldsymbol{\beta}_k\}_{k\in\mathbb{N}}$ and $\{\boldsymbol{\beta}'_k\}_{k\in\mathbb{N}}$ that share the same random samples but have different initial vectors $\boldsymbol{\beta}_0$ and $\boldsymbol{\beta}'_0$, we show in Theorem 1 that for all sufficiently small constant learning rates $\alpha$, the initialization is forgotten exponentially fast in the sense that

$$(\mathbb{E}|\boldsymbol{\beta}_k - \boldsymbol{\beta}'_k|_s^q)^{1/q} \leq r_{\alpha,s,q}^k |\boldsymbol{\beta}_0 - \boldsymbol{\beta}'_0|_s \text{ holds for all } k \in \mathbb{N}, \tag{4}$$

for contraction speed $0 \leq r_{\alpha,s,q} < 1$, and $|\cdot|_s$ the $\ell^s$-norm, that is,

$$\left|(v_1, \ldots, v_d)^\top\right|_s = \left(\sum_{i=1}^{d} |v_i|^s\right)^{1/s}, \ s \geq 1.$$

This asserts the existence of a limiting stationary distribution of $\boldsymbol{\beta}_k$ as $k \to \infty$, thereby facilitating a systematic convergence theory of SGD even in nonlinear, overparameterized models.

Building on this new framework, we provide non-asymptotic bounds for higher-order moments of the SGD error in general $\ell^s$-norms for any finite $s \geq 2$ beyond the usual $\ell^2$-norm, extendable to max-norm $\ell^\infty$ by choosing $s \approx \log(d)$. Notably, the $\ell^\infty$-norm is frequently adopted in high-dimensional sparse or structured estimation [Wainwright, 2019]. See for instance, the max-norm convergence of the Lasso and Dantzig selector [Lounici, 2008]; the pivotal method for sup-norm bounds of the square-root Lasso [Belloni et al., 2011]; and the max-norm error control for confidence intervals in high-dimensional regression problems [Javanmard and Montanari, 2013]. In stochastic-approximation (SA), Wainwright [2019] derived $\ell^\infty$-norm bounds for Q-learning with decaying learning rates; Chen et al. [2023] derived maximal concentration bounds for SA under arbitrary norms with decaying learning rates and with contraction as an assumption; Agarwal et al. [2012] considered high-dimensional SA for strongly convex objectives with a sparse optimum, but using decaying learning rates and restricting the tails of stochastic gradients to be sub-Gaussian. To date, all the existing results are restricted to low-dimensional settings or decaying learning rates and do not carry over to overparameterized models with constant learning rates. To address this gap, we derive a sharp *high-dimensional moment inequality* (see Lemma 2) valid for a broad class of learning problems, delivering explicit non-asymptotic bounds of $\mathbb{E}|\boldsymbol{\beta}_k - \boldsymbol{\beta}^*|_s^q$ and its ASGD variant for any $q, s \geq 2$ with mild conditions, together with matching complexity guarantees, i.e., given some target error $\varepsilon > 0$ (see Proposition 2), the required number of iterations $k$ such that

$$\mathbb{E}|\bar{\boldsymbol{\beta}}_k - \boldsymbol{\beta}^*|_s^q \leq \varepsilon.$$

Although moment bounds capture average-case performance, a single execution of (A)SGD in practice demands high-probability guarantees [Valiant, 1984, Vapnik, 2000, Bach and Moulines, 2013, Durmus et al., 2021, Zhong et al., 2024]. Recent advances include a generic high-probability framework for both convex and nonconvex SGD with sub-Gaussian gradient noises [Liu et al., 2023], high-probability rates for clipped-SGD with heavy-tailed noises [Nguyen et al., 2023], and high-probability guarantees for nonconvex stochastic approximation via robust gradient clipping [Li and Liu, 2022]. However, these established high-probability bounds focus again on decaying learning rates and low dimension. Moreover, early work primarily addressed light-tailed noises where the gradients are bounded or have exponential-type moments [Nemirovski et al., 2009, Rakhlin et al., 2012, Ghadimi and Lan, 2013, Cardot et al., 2017, Harvey et al., 2019, Mou et al., 2020, Chen et al., 2023]. For the cases that only admit a polynomial tail with finite $q$-th moment, Lou et al. [2022] were the first to derive a Nagaev–type inequality [Nagaev, 1979] for low-dimensional ASGD. The rate was shown to be optimal but their bound heavily relies on the linearity of gradients and is only suitable for decaying learning rates. By leveraging a dependency-adjusted *functional dependence measure* in high-dimensional time series [Zhang and Wu, 2017], we derive a high-probability concentration bounds for high-dimensional ASGD with constant learning rates. Given a tolerance level $\delta \in (0, 1)$ and a target error $\varepsilon > 0$, we provide bounds for the required number of iterations $k$ to guarantee that

$$\mathbb{P}\big(|\bar{\boldsymbol{\beta}}_k - \boldsymbol{\beta}^*|_s \leq \varepsilon\big) \geq 1 - \delta.$$

This tail-decay result (see Eq. (10)) is proved via a new Fuk-Nagaev-type inequality (see Theorem 4) and complements our moment and complexity characterizations of large-step stochastic optimization.

## 1.1   Our Contributions

This paper contributes to theoretical advancements for understanding constant learning-rate SGD and its averaged variant (ASGD) in the challenging high-dimensional regime. Our main technical innovations and results include:

**(1) Handling Constant Learning Rates in High Dimensions.** In practice, large-scale machine learning models commonly deploy fixed, large learning rates to speed up optimization in high-dimensional settings. To address this, we introduce novel coupling techniques inspired by high-dimensional nonlinear time series and establish the asymptotic stationarity of the SGD iterates with arbitrary initialization *(Section 2)*.

**(2) Generalized Moment Convergence in $\ell^s$- and $\ell^\infty$-Norms.** By deriving a sharp high-dimensional moment inequality, we establish explicit, non-asymptotic $q$-th moment bounds for arbitrary $\ell^s$-norms of (A)SGD iterates for any $q \geq 2$ and even integers $s$, generalizing previous theory primarily focusing on mean squared error (MSE) convergence with $q = s = 2$. Our results extend naturally to the max-norm case (i.e., $\ell^\infty$) by selecting $s \approx \log(d)$, that is essential for modern sparse and structured estimation in high-dimensional data *(Section 3)*.

**(3) High-Probability Tail Bounds.** While average-case (moment) bounds are informative, single runs require tail guarantees. We derive the first high-probability concentration bounds for ASGD in high-dimensional settings with constant learning rates. By developing a tight Fuk-Nagaev-type inequality using the coupling techniques in nonlinear time series, we control the algorithmic complexity required to achieve targeted accuracy with high confidence *(Section 4)*.

## 1.2   Related Works

*Stochastic Gradient Descent and its Variants.* The SGD algorithm can be traced back to Robbins and Monro [1951], Kiefer and Wolfowitz [1952]. Popular SGD variants include Nesterov's accelerated gradient [Nesterov, 1983], AdaGrad [Duchi et al., 2011], AdaDelta [Zeiler, 2012], Adam [Kingma and Ba, 2014], AMSGrad [Reddi et al., 2018], AdamW [Loshchilov and Hutter, 2018], SAG [Schmidt et al., 2017], SVRG [Johnson and Zhang, 2013], SARAH [Nguyen et al., 2017], SPIDER [Fang et al., 2018] and Katyusha [Allen-Zhu, 2017]. The theoretical foundations of SGD under decaying learning rates were established in the early studies by [Blum, 1954, Dvoretzky, 1956, Sacks, 1958], with stronger almost-sure guarantees by Fabian [1968], Robbins and Siegmund [1971], Ljung [1977], Lai [2003], Wang and Gao [2010]. Existing works for smooth, strongly-convex objectives with decaying step sizes include Ruppert [1988], Polyak and Juditsky [1992], Nemirovski et al. [2009], Bach and Moulines [2013], Rakhlin et al. [2012], Mertikopoulos et al. [2020] among others. Despite the rich literature on SGD, the theoretical understanding in high-dimensional settings remains limited.

Exceptions are Paquette et al. [2021, 2022] who study high-dimensional SGD for the least-squares loss.

*Constant Learning Rate.* In high-dimensional scenarios, constant learning rates prevail due to simpler tuning procedures and faster convergence [Wang et al., 2022]. More recent theoretical and empirical studies of large-step SGD include Wu et al. [2018], Cohen et al. [2021] and the very recent Cai et al. [2024], which formalize the resurgence of constant-step methods in modern machine learning. A useful way to analyze constant–step SGD is to treat its iterates as a time-homogeneous Markov chain [Pflug, 1986], which makes it possible to characterize its long-run behavior and stationary law. However, previous works only derived convergence in Wasserstein distance [Dieuleveut et al., 2020, Merad and Gaïffas, 2023]. Such convergence in probability measures can hardly provide refined (non)-asymptotics such as higher-moment convergence and concentration inequalities, and seems nontrivial to extend to high-dimensional regimes.

*High-Dimensional Nonlinear Time Series.* An alternative approach for constant learning-rate SGD is to view it as an iterated random function [Dubins and Freedman, 1966, Barnsley and Demko, 1985, Diaconis and Freedman, 1999, Diaconis and Duflo, 2000], or a nonlinear autoregressive (AR) process. This interpretation facilitates the theory of online learning with non-stationarity and complex dependency structures; see, for example, the recent work by Li et al. [2024c] on SGD with dropout regularization building on the GMC framework [Wu and Shao, 2004]. To extend this systematic theory to high-dimensional settings, we adapt the coupling techniques in time series [Wu, 2005, 2007, 2009, 2011, Xiao and Wu, 2012, Berkes et al., 2014, Wu and Wu, 2016, Karmakar and Wu, 2020], especially the ones for high-dimensional regimes [Zhang and Wu, 2017, 2021, Li et al., 2024a] to online learning algorithms.

## 1.3 Notation

Denote column vectors in $\mathbb{R}^d$ by lowercase bold letters $\boldsymbol{x} = (x_1, \ldots, x_d)^\top$ and the $\ell^s$-norm of $\boldsymbol{x}$ by $|\boldsymbol{x}|_s = (\sum_{i=1}^d |x_i|^s)^{1/s}$, $s \geq 1$. Write $\boldsymbol{x}^{\odot s} = (x_1^s, \ldots, x_d^s)^\top$. The expectation and covariance of random vectors are respectively denoted by $\mathbb{E}[\cdot]$ and $\text{Cov}(\cdot)$. For $q > 0$ and a random variable $X$, we write $X \in \mathcal{L}^q$ iff $\|X\|_q = [\mathbb{E}(|X|^q)]^{1/q} < \infty$. We denote matrices by uppercase letters. Given matrices $A$ and $B$ of compatible dimension, their matrix product is denoted by juxtaposition. Write $A^\top$ for the transpose of $A$ and $I_d$ for $d \times d$ identity matrix. For two positive number sequences $(a_n)$ and $(b_n)$, we say $a_n = O(b_n)$ (resp. $a_n \asymp b_n$) if there exists $c > 0$ such that $a_n/b_n \leq c$ (resp. $1/c \leq a_n/b_n \leq c$) for all large $n$. Let $(x_n)$ and $(y_n)$ be two sequences of random variables. Write $x_n = O_\mathbb{P}(y_n)$ if for $\forall \epsilon > 0$, there exists $c > 0$ such that $\mathbb{P}(|x_n/y_n| \leq c) > 1 - \epsilon$ for all large $n$.

| Notation | Definition | Reference | Index Range |
|---|---|---|---|
| $\boldsymbol{\beta}^*$ | minimizer of the loss function $G(\boldsymbol{\beta})$ | Eq. (1) | / |
| $\boldsymbol{\beta}_k$ | SGD iterates | Eq. (2) | $k \in \mathbb{N}$ |
| $\boldsymbol{\beta}_k^\circ$ | stationary SGD iterates | Thm. (2) | $k \in \mathbb{Z}$ |
| $\bar{\boldsymbol{\beta}}_k$ | ASGD iterates | Eq. (3) | $k \in \mathbb{N}$ |
| $\bar{\boldsymbol{\beta}}_k^\circ$ | stationary ASGD iterates | Eq. (9) | $k \in \mathbb{Z}$ |

Table 1: List of the sequences defined in the paper.

## 2   Convergence of SGD to a Stationary Distribution

In this section, we establish the GMC property of high-dimensional SGD with constant learning rates. Our technique is to construct a smooth surrogate for the non-differentiable $\ell^\infty$-norm via the $\ell^s$-norm, so that standard gradient-based tools become available. We defer the technical details to Section 6.1. Furthermore, we provide a novel high-dimensional moment inequality (see Section 6.2) and use it to derive the dimension-dependent range of the constant learning rate that guarantees the contraction.

We first impose the following assumptions on the objective function and the stochastic gradients.

**Assumption 1** (Coercivity). *Assume that for any sequence $\boldsymbol{\beta}_1, \boldsymbol{\beta}_2, \ldots$ with $|\boldsymbol{\beta}_n|_s \to \infty$ the loss function $G(\cdot)$ in (1) satisfies $\lim_{n \to \infty} G(\boldsymbol{\beta}_n) = \infty$.*

**Assumption 2** (Strong Convexity – $\ell^s$-norm). *Let $s \geq 2$ be an even integer and write $\boldsymbol{v}^{\odot s} := (v_1^s, \ldots, v_d^s)^\top$ for a vector $\boldsymbol{v} = (v_1, \ldots, v_d)^\top$. Assume there exists $\mu > 0$ such that*

$$\left\langle (\boldsymbol{\beta} - \boldsymbol{\beta}')^{\odot(s-1)}, \nabla G(\boldsymbol{\beta}) - \nabla G(\boldsymbol{\beta}') \right\rangle \geq \mu |\boldsymbol{\beta} - \boldsymbol{\beta}'|_s^s, \quad \text{for all } \boldsymbol{\beta}, \boldsymbol{\beta}' \in \mathbb{R}^d.$$

In Lemma 3 in the supplementary materials, we show that under Assumptions 1 and 2, a unique global minimizer $\boldsymbol{\beta}^*$ exists for the optimization problem (1). When $s = 2$, Assumption 2 reduces to the regular strong convexity frequently adopted in the literature [Polyak and Juditsky, 1992, Moulines and Bach, 2011, Dieuleveut et al., 2020, Mies and Steland, 2023]. For general $s$ and the linear regression model, Section 8.2 in the supplementary material interprets the $\ell^s$-type strong convexity assumption via the $\ell^s$-norm induced matrix norm. As different norms are involved, there does not seem to be an apparent relationship between the classical strong convexity and the case $s > 2$.

**Assumption 3** (Stochastic Lipschitz Continuity – $\ell^s$-norm). *Let $\boldsymbol{\beta}^*$ be the global minimizer. For some $q \geq 2$ and an even integer $s \geq 2$, assume that*

$$M_{s,q} := \left( \mathbb{E} |\nabla g(\boldsymbol{\beta}^*, \boldsymbol{\xi})|_s^q \right)^{1/q} < \infty.$$

*Further assume there exists a constant $L_{s,q} > 0$ such that*

$$\left( \mathbb{E} \big| \nabla g(\boldsymbol{\beta}, \boldsymbol{\xi}) - \nabla g(\boldsymbol{\beta}', \boldsymbol{\xi}) \big|_s^q \right)^{1/q} \leq L_{s,q} |\boldsymbol{\beta} - \boldsymbol{\beta}'|_s, \quad \text{for all } \boldsymbol{\beta}, \boldsymbol{\beta}' \in \mathbb{R}^d.$$

Later we will choose $s = O(\log(d))$ to bound the max-norm. The above defined Lipschitz constant $L_{s,q}$ and the moments $M_{s,q}$ will then grow as $d$ increases. Taking linear regression as an example, we investigate the dimension dependence of $L_{s,q}$ and $M_{s,q}$ in Section 8.2. All bounds in this work will contain the explicit dependence on $(L_{s,q}, M_{s,q})$.

We now state the first main result of this paper, which plays a crucial role in establishing moment convergence and tail probability results in the following sections. The statement quantifies the exponential rate at which the initialization $\boldsymbol{\beta}_0$ will be forgotten and the SGD iterates $\boldsymbol{\beta}_k$ converges to a stationary distribution $\pi_\alpha$.

**Theorem 1** (Convergence of SGD to stationary distribution). *Suppose that Assumptions 1–3 hold for some $\mu > 0$, $q \geq 2$ and even integer $s \geq 2$. Given a constant learning rate*

$$0 < \alpha < \alpha_{s,q} := \frac{2\mu}{\max\{q, s\} L_{s,q}^2}, \tag{5}$$

*for any two $d$-dimensional SGD sequences $\{\boldsymbol{\beta}_k(\alpha)\}_{k \in \mathbb{N}}$ and $\{\boldsymbol{\beta}_k'(\alpha)\}_{k \in \mathbb{N}}$ sharing the same i.i.d. noise injections $\{\boldsymbol{\xi}_k\}_{k \geq 1}$ but possibly different initializations $\boldsymbol{\beta}_0, \boldsymbol{\beta}_0' \in \mathbb{R}^d$, the geometric-moment contraction (GMC)*

$$\big\| |\boldsymbol{\beta}_k - \boldsymbol{\beta}_k'|_s \big\|_q \leq r_{\alpha,s,q}^k |\boldsymbol{\beta}_0 - \boldsymbol{\beta}_0'|_s, \quad \text{for all } k = 0, 1, \ldots$$

*holds with contraction constant*

$$r_{\alpha,s,q} = 1 - 2\mu\alpha + \max\{q, s\} L_{s,q}^2 \alpha^2 < 1. \tag{6}$$

*Moreover, there exists a unique stationary distribution $\pi_\alpha$ with a finite $q$-th moment, that is, $\int |\boldsymbol{u}|_s^q \pi_\alpha(d\boldsymbol{u}) < \infty$, such that*

$$\boldsymbol{\beta}_k \Rightarrow \pi_\alpha, \quad \text{as } k \to \infty.$$

*Equivalently, for any continuous function $f \in \mathcal{C}(\mathbb{R}^d)$ with $|f|_\infty < \infty$,*

$$\mathbb{E}\big[ f(\boldsymbol{\beta}_k) \big] \to \int f(\boldsymbol{u}) \pi_\alpha(d\boldsymbol{u}), \quad \text{as } k \to \infty.$$

The result generalizes Li et al. [2024b] to large dimension $d$ and extends the $\ell^2$-type GMC based on Lemma 9 to general $\ell^s$-norms. Moreover, choosing $s = s_d$ with

$$s_d := 2 \min\{\ell \in \mathbb{N} : 2\ell > \log(d)\}, \tag{7}$$

and using the inequality

$$|\boldsymbol{x}|_\infty \leq |\boldsymbol{x}|_{s_d} \leq d^{1/s_d} |\boldsymbol{x}|_\infty \leq e|\boldsymbol{x}|_\infty, \tag{8}$$

shows the equivalence of the $\ell^{s_d}$- and $\ell^\infty$-norms. Consequently, by choosing $s = s_d$, the previous theorem can also be used to derive the GMC property with respect to the $\ell^\infty$-norm.

# 3 Convergence of High-Dimensional SGD and ASGD

In this section, we derive convergence rates for the moments of the last iterate $\mathbb{E}|\boldsymbol{\beta}_k - \boldsymbol{\beta}^*|_\infty^q$ and the moments of the averaged SGD.

## 3.1 Convergence of SGD

**Proposition 1.** *If Assumptions 1–3 hold for some $q \geq 2$, an even integer $s \geq 2$, and a constant $M_{s,q}$, then,*

$$\left\| |\boldsymbol{\beta}_k - \boldsymbol{\beta}^*|_s \right\|_q^2 \leq \left(1 - 2\alpha\mu + 7\max\{q,s\}\alpha^2 L_{s,q}^2\right)\left\| |\boldsymbol{\beta}_{k-1} - \boldsymbol{\beta}^*|_s \right\|_q^2 + 3\max\{q,s\}\alpha^2 M_{s,q}^2,$$

*for all $k \geq 1$. The same inequality holds if $\boldsymbol{\beta}_k$ is replaced by the stationary SGD iterates $\boldsymbol{\beta}_k^\circ \sim \pi_\alpha$, $k \geq 1$.*

**Theorem 2** (Moment convergence of SGD). *Let $0 < \alpha < \alpha_{s,q}/7$ with $\alpha_{s,q}$ as defined in (5). Suppose that Assumptions 1–3 hold for $q \geq 2$ and even integer $s \geq 2$. Then for the stationary SGD iterates $\boldsymbol{\beta}_k^\circ \sim \pi_\alpha$,*

$$\left\| |\boldsymbol{\beta}_k^\circ - \boldsymbol{\beta}^*|_s \right\|_q = O\left(M_{s,q}\sqrt{\max\{q,s\}\alpha}\right) \quad \text{for all } k \geq 1$$

*and for the SGD iterate $\boldsymbol{\beta}_k$ with arbitrary initialization $\boldsymbol{\beta}_0$,*

$$\left\| |\boldsymbol{\beta}_k - \boldsymbol{\beta}^*|_s \right\|_q = O\left(M_{s,q}\sqrt{\max\{q,s\}\alpha} + r_{\alpha,s,q}^k \| |\boldsymbol{\beta}_0 - \boldsymbol{\beta}_0^\circ|_s \|_q\right) \quad \text{for all } k \geq 1.$$

Choosing $s = s_d$ in (7) yields a bound with respect to the $\ell^\infty$-norm.

## 3.2 Convergence of Ruppert-Polyak Averaged SGD

Consider now the Ruppert-Polyak Averaged SGD (ASGD) $\bar{\boldsymbol{\beta}}_k = \frac{1}{k}\sum_{i=1}^k \boldsymbol{\beta}_i$ as defined in (3). For the initialization $\boldsymbol{\beta}_0^\circ \sim \pi_\alpha$, define the stationary ASGD sequence

$$\bar{\boldsymbol{\beta}}_k^\circ = \frac{1}{k}\sum_{i=1}^k \boldsymbol{\beta}_i^\circ, \quad k \geq 1. \tag{9}$$

**Theorem 3.** *Consider the ASGD sequence $\{\bar{\boldsymbol{\beta}}_k\}_{k\geq 1}$. Suppose that Assumptions 1–3 hold with some $q \geq 2$ and even integer $s = s_d$ in (7), the conditions of Theorem 8 hold and the learning rate satisfies $\alpha \in (0, \alpha_{s_d,q})$ with $\alpha_{s_d,q}$ defined in (5). For any $k \geq 1$ and some universal constants $C_1, C_2, C_3 > 0$,*

$$\left\| |\bar{\boldsymbol{\beta}}_k - \boldsymbol{\beta}^*|_\infty \right\|_q \leq C_1 \left\{ \underbrace{\sqrt{\frac{c_q s_d}{k}} M_{s_d,q}\left(L_{s_d,q}\sqrt{\alpha\max\{q,s_d\}} + 1\right)}_{\text{stochastic variance}} \right\}$$

$$+ C_2 \left\{ \underbrace{\frac{1}{k(1 - r_{\alpha,s_d,q})} \| |\boldsymbol{\beta}_0 - \boldsymbol{\beta}_0^\circ|_\infty \|_q}_{\text{initialization bias}} \right\} + C_3 \left\{ \underbrace{M_{s_d,q}^2 \max\{q,s_d\}\alpha d^{\frac{q}{q-1}\cdot\left(1-\frac{2}{s_d}\right)}}_{\text{bias of constant learning rate}} \right\}.$$

**Proposition 2** (Complexity bound). *Under the assumptions of Theorem 3, let $\Delta_0 = \| |\boldsymbol{\beta}_0 - \boldsymbol{\beta}_0^\circ|_\infty \|_q$,*

$$V = L_{s_d,q} M_{s_d,q}\sqrt{\max\{q,s_d\}} + M_{s_d,q}, \quad B = M_{s_d,q}^2 \max\{q,s_d\} d^{\frac{q}{q-1}\left(1-\frac{2}{s_d}\right)}.$$

*Given a tolerance $\varepsilon > 0$,*

$$\alpha \leq \min\left\{\frac{\varepsilon}{3C_3 B}, \frac{\alpha_{s_d,q}}{7}\right\}, \quad \text{and} \quad k \geq \max\left\{\frac{9C_1^2 c_q s_d V^2 \alpha}{\varepsilon^2}, \frac{3C_2 \Delta_0}{\alpha\varepsilon}\right\},$$

*we have $\| |\bar{\boldsymbol{\beta}}_k - \boldsymbol{\beta}^*|_\infty \|_q \leq \varepsilon$.*

A proof outline is given in Section 6.3 and the full proof is deferred to the supplementary material. The sharpest complexity bound of SA for $\ell^\infty$-norm known to date was derived by Wainwright [2019] proving that the number of iterations required to obtain an $\varepsilon$-accurate solution of Q-learning scales as $(1-\gamma)^{-4} \cdot \varepsilon^{-2}$ with the discount factor $\gamma$. In Proposition 2, our complexity bound for SGD is also of the order of $O(1/\varepsilon^2)$ if the dimension $d$ is fixed, which is consistent with the degenerate Q-learning case in Wainwright [2019]. The derived result allows to determine the dependence on the dimension $d$.

# 4 Sharp Concentration and Gaussian Approximation

Via the following tail probability inequality for the averaged SGD estimator $\bar{\boldsymbol{\beta}}_k$, one can further derive high-probability concentration bound of $|\bar{\boldsymbol{\beta}}_k - \boldsymbol{\beta}^*|_\infty$. Recall that $s_d = 2\min\{\ell \in \mathbb{N} : 2\ell > \log(d)\}$.

**Theorem 4** (Fuk-Nagaev inequality). *Under the conditions of Theorem 3, for any $z > 0$, we have*

$$\mathbb{P}\big(|\bar{\boldsymbol{\beta}}_k - \boldsymbol{\beta}^*|_\infty > z\big) \lesssim \frac{\||\boldsymbol{\beta}_0 - \boldsymbol{\beta}_0^\circ|_\infty\|_q^q}{(k\alpha z)^q} + \frac{(\log d)^{\frac{3q}{2}}(\log k)^{1+2q}M_{s_d,q}^q}{z^q k^{q-1}\alpha^{q/2-1}} + \exp\left(-\frac{Ckz^2\alpha^{1-2/q}}{M_{s_d,q}^2\log d}\right),$$

*where the constants in $\lesssim$ are independent of $k, d, s$ and $\alpha$.*

As an immediate consequence of Theorem 4, we obtain a sharp high-probability upper bound for $|\bar{\boldsymbol{\beta}}_k - \boldsymbol{\beta}^*|_\infty$, that is, for any given tolerance rate $\delta \in (0, 1)$, with at least probability $1 - \delta$, we have

$$|\bar{\boldsymbol{\beta}}_k - \boldsymbol{\beta}^*|_\infty = O\left(\frac{\||\boldsymbol{\beta}_0 - \boldsymbol{\beta}_0^\circ|_\infty\|_q^q}{k\alpha\delta^{1/q}} + \frac{(\log d)^{3/2}(\log k)^{1/q+2}M_{s_d,q}}{k^{1-1/q}\alpha^{1/2-1/q}\delta^{1/q}} + \sqrt{\frac{M_{s_d,q}^2\log d\log(1/\delta)}{k\alpha^{1-2/q}}}\right). \tag{10}$$

Notably, if the $q$-th moment of the gradient noise is finite ($M_{s_d,q} < \infty$), the second term of the right hand side, involving $k^{1-q}$, is generally unimprovable [Nagaev, 1979, Lou et al., 2022].

The distribution convergence for the high-dimensional ASGD relies on the following result. Let $M_{2,q}$ be as defined in Assumption 3.

**Theorem 5** (Gaussian approximation). *Consider stationary SGD iterates $\boldsymbol{\beta}_k^\circ \sim \pi_\alpha$ with $\pi_\alpha$ as defined in Theorem 1, initialization $\boldsymbol{\beta}_0^\circ \sim \pi_\alpha$, and learning rate $\alpha \in (0, \alpha_{s_d,q})$. Suppose that Assumptions 1–3 hold for some $q > 2$. Then, on a potentially different probability space, and for a number of iterations $T$ satisfying $d \leq cT$, where $c > 0$ is some constant, there exist random vectors $\{\tilde{\boldsymbol{\beta}}_k\}_{k=1}^T \overset{\mathcal{D}}{=} \{\boldsymbol{\beta}_k^\circ\}_{k=1}^T$ and independent Gaussian random vectors $\{\boldsymbol{z}_k\}_{k=1}^T$ with mean zero and covariance matrix*

$$\Xi = \sum_{k=-\infty}^{\infty} \text{Cov}(\boldsymbol{\beta}_0^\circ, \boldsymbol{\beta}_k^\circ), \tag{11}$$

*such that*

$$\left(\mathbb{E}\max_{k\leq T}\Big|\frac{1}{\sqrt{k}}\sum_{i=1}^{k}\big[(\tilde{\boldsymbol{\beta}}_i - \mathbb{E}[\boldsymbol{\beta}_1^\circ]) - \boldsymbol{z}_i\big]\Big|_2^2\right)^{1/2} \leq C_{\alpha,q}^* M_{2,q}\sqrt{d\log(T)}\Big(\frac{d}{T}\Big)^{\frac{q-2}{6q-4}}, \tag{12}$$

*with $C_{\alpha,q}^*$ a constant that only depends on $c$, the learning rate $\alpha$, and the moment index $q$.*

For diverging moment index $q \to \infty$, the Gaussian approximation rate in (12) approaches the rate $O(\sqrt{\log(T)}(d^4/T)^{1/6})$. Thus, to obtain a nontrivial Gaussian approximation bound within $T$ iterations, we need dimension dependence $d = o(T^{1/4-\zeta})$ with $\zeta > 0$.

# 5 Constant Learning Rate for Large Dimension

Recall that $L_{s,q}$ is the Lipschitz constant introduced in Assumption 3. We established asymptotic stationarity and non-asymptotic convergence if $\alpha < \alpha_{s,q}/7$ with $\alpha_{s,q}$ defined in (5), leading to the upper bound

$$\alpha < \frac{\alpha_{s,q}}{7} = \frac{2\mu}{7\max\{q,s\}L_{s,q}^2} \asymp \frac{1}{d^2\log(d)},$$

if we choose $s = s_d$ in (7) and if $L_{s_d,q} \asymp d$. We refer to Section 8.2 for the derivation of the dimension dependence of $L_{s,q}$ in the linear regression model.

Alternatively, the upper bound for the learning rate $\alpha$ can also be derived by a linear approximation technique (see Lemma 1), defined as the nontrivial solution to the following equation

$$1 - q\mu\alpha + \frac{q\big[|q-s| + (s-1)\big]L_{s,q}^2}{2}\alpha^2(1 + \alpha L_{s,q})^{q-2} = 1. \tag{13}$$

A derivation of this equation is provided in Section 6.1. The existence of a solution of (13) is shown below the proof of Lemma 1 in the supplementary materials. When $q = 2$, the range of $\alpha$ simplifies to

$$\alpha < \frac{2\mu}{7\big[|s-2| + (s-1)\big]L_{s,2}^2},$$

which is also proportional to $1/[d^2 \log(d)]$ if we choose $s = s_d$ in (7) and if $L_{s_d,2} \asymp d$, matching the rate of $\alpha_{s,q}$ in (5) derived by Lemma 2, though with a slightly more conservative constant for general $s$. In the special case with $s = 2$, both bounds reduce to the classical $\alpha < 2\mu/L_{2,2}^2$. If $L_{2,2} \asymp d$ for large dimension $d$, which is shown to be true for the linear regression model in Section 8.2 in the supplementary materials, the $\ell^\infty$- and the $\ell^2$-norm yield similar upper bounds for the learning rate $\alpha$.

# 6 Proof Sketches

## 6.1 Bridge between $\ell^s$- and $\ell^\infty$- Norms

In high-dimensional regimes, convergence rates of constant-learning-rate SGD (2) with respect to the $\ell^\infty$-norm are of particular interest [Wainwright, 2019, Chen et al., 2023]. However, it is extremely challenging to directly study the convergence of $|\boldsymbol{\beta}_k - \boldsymbol{\beta}^*|_\infty$ since the $\ell^\infty$-norm is not differentiable thereby ruling out standard gradient-based tools for proving convergence rates or concentration. To address this issue, we instead study $|\cdot|_{s_d}$ with $s_d$ defined in (7). By the equivalence between $\ell^{s_d}$- and $\ell^\infty$-norms shown in (8), contraction in $\ell^\infty$-norm follows from $\ell^{s_d}$-norm contraction.

To prove the GMC property of SGD as introduced in (4), it suffices to show that for any two $d$-dimensional SGD sequences $\{\boldsymbol{\beta}_k\}_{k\in\mathbb{N}}$ and $\{\boldsymbol{\beta}'_k\}_{k\in\mathbb{N}}$ sharing the same i.i.d. observations $\{\boldsymbol{\xi}_k\}_{k\geq 1}$ but possibly different initializations $\boldsymbol{\beta}_0, \boldsymbol{\beta}'_0 \in \mathbb{R}^d$, the contraction holds for $|\boldsymbol{\beta}_k - \boldsymbol{\beta}'_k|_{s_d}$ for all $k \geq 1$. To this end, we need to determine a range of constant learning rates $\alpha$ such that for any $q \geq 2$ and $\boldsymbol{\beta}, \boldsymbol{\beta}' \in \mathbb{R}^d$, the GMC in Theorem 1 holds, i.e.,

$$\big(\mathbb{E}\big|\boldsymbol{\beta} - \alpha\nabla g(\boldsymbol{\beta}, \boldsymbol{\xi}) - \big(\boldsymbol{\beta}' - \alpha\nabla g(\boldsymbol{\beta}', \boldsymbol{\xi})\big)\big|_s^q\big)^{1/q} \leq r|\boldsymbol{\beta} - \boldsymbol{\beta}'|_s, \quad \text{for some } r = r_{\alpha,s,q} < 1. \quad (14)$$

To derive the inequality, we first provide a lemma based on linear approximation by considering the scalar function

$$\alpha \mapsto |\boldsymbol{x} - \alpha\boldsymbol{z}|_s^q, \quad \text{where } \boldsymbol{x} = \boldsymbol{\beta} - \boldsymbol{\beta}', \quad \boldsymbol{z} = \nabla g(\boldsymbol{\beta}, \boldsymbol{\xi}) - \nabla g(\boldsymbol{\beta}', \boldsymbol{\xi}),$$

and linearizing it around $\alpha = 0$. Then, one only needs to prove that $\mathbb{E}|\boldsymbol{x} - \alpha\boldsymbol{z}|_s^q \leq r|\boldsymbol{x}|_s^q$. By the second-order Taylor expansion of $|\boldsymbol{x} - \alpha\boldsymbol{z}|_s^q$ in $\alpha$, we have the linear approximation

$$|\boldsymbol{x} - \alpha\boldsymbol{z}|_s^q \approx |\boldsymbol{x}|_s^q - q\alpha|\boldsymbol{x}|_s^{q-s}\langle\boldsymbol{x}^{s-1}, \boldsymbol{z}\rangle, \quad (15)$$

with remainder term of order $\alpha^2$, see Section 2 in the supplementary materials for details. Since a simple triangle inequality argument $\||\boldsymbol{x} - \alpha\boldsymbol{z}|_s\|_q \leq \||\boldsymbol{x}|_s\|_q + \alpha\||\boldsymbol{z}|_s\|_q$ fails to control this remainder sufficiently to yield a contraction constant $r < 1$, we establish a more precise bound.

**Lemma 1.** *Recall that $\boldsymbol{v}^{\otimes s} = (v_1^s, \ldots, v_d^s)^\top$ for a vector $\boldsymbol{v} = (v_1, \ldots, v_d)^\top$. For any $q \geq 2$, any even integer $s \geq 2$, any two vectors $\boldsymbol{x}, \boldsymbol{z} \in \mathbb{R}^d$, and any $\alpha > 0$,*

$$\Big||\boldsymbol{x} - \alpha\boldsymbol{z}|_s^q - |\boldsymbol{x}|_s^q + q\alpha|\boldsymbol{x}|_s^{q-s}\langle\boldsymbol{x}^{s-1}, \boldsymbol{z}\rangle\Big| \leq \frac{q\alpha^2}{2}\big[|q-s| + (s-1)\big]\big(|\boldsymbol{x}|_s + \alpha|\boldsymbol{z}|_s\big)^{q-2}|\boldsymbol{z}|_s^2.$$

If $s = 2, q = 2$, the right-hand side is $\alpha^2|\boldsymbol{z}|_2^2$. This is consistent with the Taylor remainder of the right-hand side in Lemma 9 derived by Li et al. [2024c]. Using this inequality to prove the contraction in (14) is remarkably different from the approaches relying on the martingale decomposition (MD) that is frequently adopted in the literature [Dieuleveut et al., 2020, Mertikopoulos et al., 2020, Mies and Steland, 2023]. Our proposed method requires mild moment conditions on the stochastic gradients and yields simpler proofs that can be generalized to a broad class of online learning problems. We refer to Li et al. [2024b] for detailed discussion. Nevertheless, we remark in advance that a Rio-type inequality (Lemma 2) with slightly sharper constants will be used directly in our main contraction proof, while we retain Lemma 1 here for its intuitive appeal. Finally, by choosing $s = s_d$ as in (7) for (14), we can expect the $\ell^\infty$-norm type GMC to hold for high-dimensional SGD iterates.

## 6.2 High-Dimensional Moment Inequality

To prove Theorem 1, we derive a high-dimensional version of Rio's inequality [Rio, 2009], adapted to the $q$-th moment of $\ell^s$-norm. This result provides a slightly sharper constant than Lemma 1 and is used directly in our moment-contraction analysis.

**Lemma 2** (High-dimensional moment inequality). *For any $q \geq 2$, any even integer $s \geq 2$, and any two $d$-dimensional random vectors $\boldsymbol{x}, \boldsymbol{y}$, we have*

$$\big\| |\boldsymbol{x} + \boldsymbol{y}|_s \big\|_q^2 \leq \big\| |\boldsymbol{x}|_s \big\|_q^2 + 2 \big\| |\boldsymbol{x}|_s \big\|_q^{2-q} \mathbb{E}\Big( |\boldsymbol{x}|_s^{q-s} \sum_{j=1}^{d} x_j^{s-1} y_j \Big) + \big( \max\{q,s\} - 1 \big) \big\| |\boldsymbol{y}|_s \big\|_q^2.$$

*Moreover, if $\mathbb{E}[\boldsymbol{y} \mid \boldsymbol{x}] = 0$ almost surely, then*

$$\big\| |\boldsymbol{x} + \boldsymbol{y}|_s \big\|_q^2 \leq \big\| |\boldsymbol{x}|_s \big\|_q^2 + \big( \max\{q,s\} - 1 \big) \big\| |\boldsymbol{y}|_s \big\|_q^2. \tag{16}$$

Repeatedly applying Lemma 2 leads to the high-dimensional maximal moment inequality in Lemma 8 in the supplementary materials, which is of independent interest.

## 6.3 Stationarity, Variation and Bias of ASGD

We prove the moment bound $\big\| |\bar{\boldsymbol{\beta}}_k - \boldsymbol{\beta}^*|_\infty \big\|_q$ via the decomposition

$$\big\| |\bar{\boldsymbol{\beta}}_k - \boldsymbol{\beta}^*|_\infty \big\|_q \leq \big\| |\bar{\boldsymbol{\beta}}_k - \bar{\boldsymbol{\beta}}_k^\circ|_\infty \big\|_q + \big\| |\bar{\boldsymbol{\beta}}_k^\circ - \mathbb{E}[\bar{\boldsymbol{\beta}}_k^\circ]|_\infty \big\|_q + \big| \mathbb{E}[\bar{\boldsymbol{\beta}}_k^\circ] - \boldsymbol{\beta}^* \big|_\infty.$$

The first term accounts for the deviation due to the non-stationarity of $\bar{\boldsymbol{\beta}}_k$ as it is initialized from an arbitrarily fixed $\boldsymbol{\beta}_0$; this can be bounded using the GMC property of $\boldsymbol{\beta}_k$ shown in Theorem 1. The second term captures the stochastic variance of the stationary ASGD sequence. Bounding this term is more delicate because of the intricate dependency structure of $\bar{\boldsymbol{\beta}}_k^\circ$. To address this, we deploy another powerful tool in time series – the *functional dependence measure* [Wu, 2005] in Section 8.6 of the supplementary materials, which can effectively quantify the contribution of the random sample $\boldsymbol{\xi}_i$ to the $k$-th SGD iterate $\boldsymbol{\beta}_k^\circ$ for all $i \leq k$. As such, by controlling the cumulative dependence measures, we can bound this variance. Lastly, we handle the third term, which represents the non-diminishing bias of $\bar{\boldsymbol{\beta}}_k^\circ$ induced by the constant learning rate $\alpha$ [Dieuleveut et al., 2020, Huo et al., 2023]. This can be dealt with by extending the approach in Li et al. [2024b] to high-dimensional settings.

**Theorem 6** (Asymptotic stationarity). *Consider the ASGD iterates $\bar{\boldsymbol{\beta}}_k$ and the stationary version $\bar{\boldsymbol{\beta}}_k^\circ$. Suppose that Assumptions 1–3 are satisfied for some $q \geq 2$ and some even integer $s \geq 2$. Then, for the learning rate $\alpha \in (0, \alpha_{s,q})$ with $\alpha_{s,q}$ defined in (5),*

$$\big\| |\bar{\boldsymbol{\beta}}_k - \bar{\boldsymbol{\beta}}_k^\circ|_s \big\|_q \leq \frac{1}{k} \cdot \frac{1}{1 - r_{\alpha,s,q}} |\boldsymbol{\beta}_0 - \boldsymbol{\beta}_0^\circ|_s.$$

As a direct consequence of Theorem 6, we have $\big\| |\bar{\boldsymbol{\beta}}_k - \bar{\boldsymbol{\beta}}_k^\circ|_s \big\|_q \lesssim |\boldsymbol{\beta}_0 - \boldsymbol{\beta}_0^\circ|_s / (k\alpha)$, which indicates the asymptotic stationarity of high-dimensional ASGD sequences. When the bias induced by the initialization is controlled, i.e., $|\boldsymbol{\beta}_0 - \boldsymbol{\beta}_0^\circ|_s < \infty$, as $k\alpha \to \infty$, the ASGD iterate $\bar{\boldsymbol{\beta}}_k$ approaches the stationary solution $\bar{\boldsymbol{\beta}}_k^\circ$ in the sense that $\big\| |\bar{\boldsymbol{\beta}}_k - \bar{\boldsymbol{\beta}}_k^\circ|_s \big\|_q \to 0$. By Theorem 6, we only need to show the convergence for stationary ASGD.

**Theorem 7** (Stochasticity of stationary ASGD). *Consider the stationary SGD sequence $\{\boldsymbol{\beta}_k^\circ\}_{k \geq 1}$. Suppose that Assumptions 1–3 hold with some $q \geq 2$ and some even integer $s \geq 2$. Then there exists a constant $c_q > 0$ only depending on $q$, such that, for all $k \geq 1$,*

$$\big\| |\bar{\boldsymbol{\beta}}_k^\circ - \mathbb{E}[\bar{\boldsymbol{\beta}}_k^\circ]|_s \big\|_q \leq \sqrt{\frac{c_q s}{k}} M_{s,q} \Big( L_{s,q} \sqrt{\alpha \max\{q,s\}} + 1 \Big).$$

In the low-dimensional case, we take $s = 2$ as a special example. Then, $L_{s,q} \sqrt{\alpha \max\{q,s\}} = O(1)$ such that the bound is $\big\| |\bar{\boldsymbol{\beta}}_k^\circ - \mathbb{E}[\bar{\boldsymbol{\beta}}_k^\circ]|_s \big\|_q = O\{1/\sqrt{k}\}$. This rate is optimal considering the central limit theorem of the stationary ASGD.

Next, we consider the bias induced by the constant learning rate. We first introduce some necessary notation. Recall $G(\boldsymbol{\beta}) = \mathbb{E}[\nabla g(\boldsymbol{\beta}, \boldsymbol{\xi})]$ and $\nabla G(\boldsymbol{\beta}) = (\partial_1 G(\boldsymbol{\beta}), \ldots, \partial_d G(\boldsymbol{\beta}))^\top$, where $\boldsymbol{\beta} = (\beta_1, \ldots, \beta_d)^\top$. Denote $\partial_i G(\boldsymbol{\beta}) = \partial G(\boldsymbol{\beta}) / \partial \beta_i$, $1 \leq i \leq d$,

$$\nabla^2 G(\boldsymbol{\beta}) = \big[ \partial_i \partial_j G(\boldsymbol{\beta}) \big]_{1 \leq i,j \leq d}, \quad \nabla^3 G_i(\boldsymbol{\beta}) = \big[ \partial_i \partial_l \partial_r G(\boldsymbol{\beta}) \big]_{1 \leq l,r \leq d}. \tag{17}$$

We provide the non-asymptotic bound for the bias of stationary ASGD in the following lemma.

**Theorem 8** (Bias of stationary ASGD). *Under Assumptions 1–3, consider the stationary ASGD $\bar{\boldsymbol{\beta}}_k^\circ$. Assume that $g(\boldsymbol{\beta}, \xi)$ is twice differentiable with respect to $\boldsymbol{\beta}$ with positive definite Hessian matrix $\nabla^2 G(\boldsymbol{\beta}^*)$, and uniformly bounded derivatives $\max_{1 \le i \le d} \|\nabla^3 G_i(\boldsymbol{\beta})\|_\infty < \infty$, where*

$$\|\nabla^3 G_i(\boldsymbol{\beta})\|_\infty := \max_{1 \le l \le d} \sum_{r=1}^d \left| \left( \nabla^3 G_i(\boldsymbol{\beta}) \right)_{l,r} \right|.$$

*Then, we have*

$$\left| \mathbb{E}[\bar{\boldsymbol{\beta}}_k^\circ - \boldsymbol{\beta}^*] \right|_\infty = O\left( M_{s_d,q}^2 \max\{q, s_d\} \alpha d^{\frac{q}{q-1} \cdot (1 - \frac{2}{s_d})} \right).$$

## 7 Conclusions and Discussion

This work advances the theoretical understanding of the constant learning-rate SGD algorithms in high-dimensional settings. By introducing novel coupling techniques in nonlinear time series, we establish asymptotic stationarity of SGD with any initialization. We then derive non-asymptotic $q$-th moment bounds in general $\ell^s$- and $\ell^\infty$-norms, and develop the first Fuk-Nagaev high-probability tail bound for ASGD. While this paper assumes strong convexity and smoothness of the objective, the nonlinear time series perspective offers a principled framework applicable to a broad class of over-parameterized optimization tasks and can be extended to non-convex regimes, providing fundamental insights into the stability, convergence, and reliability of large-scale learning algorithms.

## Acknowledgments and Disclosure of Funding

We sincerely thank the program chair, senior area chair, area chair, and the four reviewers for their constructive feedback and involved discussion, which has greatly improved the clarity of our paper. Jiaqi Li's research is partially supported by the NSF (Grant NSF/DMS-2515926). Johannes Schmidt-Hieber has received funding from the Dutch Research Council (NWO) via the Vidi grant VI.Vidi.192.021. Wei Biao Wu's research is partially supported by the NSF (Grants NSF/DMS-2311249, NSF/DMS-2027723). We would like to thank Insung Kong for helpful discussions.

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

# 8 Technical Appendices and Supplementary Material

## 8.1 Existence and Uniqueness of Global Minimum

**Lemma 3.** *Consider the minimization problem $\boldsymbol{\beta}^* \in \arg\min_{\boldsymbol{\beta} \in \mathbb{R}^d} G(\boldsymbol{\beta})$. If the function $G$ satisfies Assumptions 1 and 2, then a global minimizer $\boldsymbol{\beta}^*$ exists and is unique.*

*Proof of Lemma 3.* We first show the existence of a global minimizer. By the coercivity condition in Assumption 1, $\lim_{|\boldsymbol{\beta}|_s \to \infty} G(\boldsymbol{\beta}) = \infty$, which implies that we can choose some large $\delta \in \mathbb{R}$ such that the sub-level set

$$\mathcal{S}_\delta := \{\boldsymbol{\beta} \in \mathbb{R}^d : G(\boldsymbol{\beta}) \leq \delta\}$$

is non-empty and bounded. Since $G$ is continuous by Assumption 2, $\mathcal{S}_\delta$ is also closed, and hence compact in $\mathbb{R}^d$ by the Heine–Borel theorem. Finally, by applying the Weierstrass extreme value theorem, there exists $\boldsymbol{\beta}^* \in \mathcal{S}_\delta$ such that $G(\boldsymbol{\beta}^*) = \min_{\boldsymbol{\beta} \in \mathcal{S}_\delta} G(\boldsymbol{\beta})$. Since for any $\boldsymbol{\beta} \notin \mathcal{S}_\delta$, $G(\boldsymbol{\beta}) > \delta \geq G(\boldsymbol{\beta}^*)$, $G(\boldsymbol{\beta}^*) = \min_{\boldsymbol{\beta} \in \mathbb{R}^d} G(\boldsymbol{\beta})$.

Next, we show the uniqueness of the global minium. Assume that there are two distinct minimizers $\boldsymbol{\beta}_1 \neq \boldsymbol{\beta}_2$. By Assumption 2, there exists $\mu > 0$ such that

$$\langle (\boldsymbol{\beta}_1 - \boldsymbol{\beta}_2)^{\odot(s-1)}, \nabla G(\boldsymbol{\beta}_1) - \nabla G(\boldsymbol{\beta}_2) \rangle \geq \mu |\boldsymbol{\beta}_1 - \boldsymbol{\beta}_2|_s^s > 0.$$

However, since $\boldsymbol{\beta}_1$ and $\boldsymbol{\beta}_2$ are both minimizers, $\nabla G(\boldsymbol{\beta}_1) = \nabla G(\boldsymbol{\beta}_2) = 0$, while $\mu |\boldsymbol{\beta}_1 - \boldsymbol{\beta}_2|_s^s > 0$. This leads to contradiction, which finishes the proof. $\qquad\square$

## 8.2 Example: Linear Regression

As example, we consider the SGD algorithm for the high-dimensional linear regression, observing independent and identically distributed (i.i.d.) pairs $\boldsymbol{\xi}_1 := (\boldsymbol{x}_1, y_1), \boldsymbol{\xi}_2 := (\boldsymbol{x}_2, y_2), \ldots$ satisfying

$$y_k = \boldsymbol{x}_k^\top \boldsymbol{\beta} + \epsilon_k, \quad \text{for } k = 1, 2, \ldots, \tag{18}$$

for random noises $\epsilon_k$ that are independent of $\boldsymbol{x}_k$ with $\mathbb{E}[\epsilon_k] = 0$ and $\mathbb{E}|\epsilon_k|^q < \infty$ for some $q \geq 2$. We verify Assumptions 2 and 3 and derive the explicit dependency of the learning-rate, the Lipschitz constant, and the moments of the gradient noise on the dimension $d$.

Let $\boldsymbol{\xi} = (y, \boldsymbol{x})$ be an independent random sample from the same distribution as the data. The least-squares loss and the stochastic gradient are respectively given by

$$g(\boldsymbol{\beta}, \boldsymbol{\xi}) = \frac{1}{2}(y - \boldsymbol{x}^\top \boldsymbol{\beta})^2, \quad \text{and } \nabla g(\boldsymbol{\beta}, \boldsymbol{\xi}) = -(y - \boldsymbol{x}^\top \boldsymbol{\beta})\boldsymbol{x}. \tag{19}$$

Then

$$\nabla G(\boldsymbol{\beta}) = \mathbb{E}[\nabla g(\boldsymbol{\beta}, (y, \boldsymbol{x})] = -\mathbb{E}[(y - \boldsymbol{x}^\top \boldsymbol{\beta})\boldsymbol{x}] = \mathbb{E}[\boldsymbol{x}\boldsymbol{x}^\top](\boldsymbol{\beta} - \boldsymbol{\beta}^*). \tag{20}$$

Let

$$\Sigma = \mathbb{E}[\boldsymbol{x}\boldsymbol{x}^\top], \quad \boldsymbol{v} = \boldsymbol{\beta} - \boldsymbol{\beta}'. \tag{21}$$

To verify the $\ell^s$-type strong convexity

$$\langle (\boldsymbol{\beta}_1 - \boldsymbol{\beta}_2)^{\odot(s-1)}, \nabla G(\boldsymbol{\beta}) - \nabla G(\boldsymbol{\beta}') \rangle \geq \mu |\boldsymbol{\beta} - \boldsymbol{\beta}'|_s^s, \quad \text{for all } \boldsymbol{\beta}, \boldsymbol{\beta}' \in \mathbb{R}^d,$$

imposed in Assumption 2, observe that $\nabla G(\boldsymbol{\beta}) - \nabla G(\boldsymbol{\beta}') = \Sigma \boldsymbol{v}$. Thus, the condition becomes

$$0 < \lambda_{\min}^{(s)} := \inf_{\boldsymbol{v} \in \mathbb{R}^d, \boldsymbol{v} \neq 0} \frac{\langle \boldsymbol{v}^{s-1}, \Sigma \boldsymbol{v} \rangle}{|\boldsymbol{v}|_s^s}. \tag{22}$$

**Lemma 4.** *Let $s \in \{2, 4, 6, \ldots\}$. Writing $\Sigma = (\Sigma_{ij})_{i,j=1,\ldots,d}$, we have*

$$\lambda_{\min}^{(s)} \geq \min_{i=1,\ldots,d} \Sigma_{ii} - \sum_{j:j\neq i} |\Sigma_{ij}|.$$

*Proof of Lemma 4.* Write $\boldsymbol{v} = (v_1, \dots, v_d)^\top$. Because of $(|v_i|^{s-1} - |v_j|^{s-1})(|v_i| - |v_j|) \geq 0$, we obtain $|v_i^{s-1} v_j| + |v_i v_j^{s-1}| \leq v_i^s + v_j^s$ and

$$
\begin{aligned}
\langle \boldsymbol{v}^{\odot(s-1)}, \Sigma \boldsymbol{v} \rangle &= \sum_{i=1}^d \Sigma_{ii} v_i^s - \sum_{i<j} \Sigma_{ij}\left(v_i^{s-1}v_j + v_i v_j^{s-1}\right) \\
&\geq \sum_{i=1}^d \Sigma_{ii} v_i^s - \sum_{i<j} |\Sigma_{ij}|\left(v_i^s + v_j^s\right) \\
&\geq \left(\min_{i=1,\dots,d} \Sigma_{ii} - \sum_{j:j\neq i} |\Sigma_{ij}|\right)\sum_\ell v_\ell^s.
\end{aligned}
$$

$\square$

This shows that the rightmost "Gershgorin gap" $\min_{i=1,\dots,d} \Sigma_{ii} - \sum_{j:j\neq i} |\Sigma_{ij}|$ is a universal lower bound for every $s$. The lower bound is non-trivial if $\Sigma$ is sufficiently diagonally dominant.

For large $s$, the inequality $\lambda_{\min}^{(s)} \geq \min_{i=1,\dots,d} \Sigma_{ii} - \sum_{j:j\neq i} |\Sigma_{ij}|$ is nearly sharp. To see this, let $i^*$ be the index $i$ that minimizes $\min_{i=1,\dots,d} \Sigma_{ii} - \sum_{j:j\neq i} |\Sigma_{ij}|$. For a small $\delta > 0$, pick $\boldsymbol{v} = (v_1, \dots, v_d)$ by choosing $v_{i^*} := 1$ and for $i \neq i^*$, taking $v_i := -\operatorname{sign}(\Sigma_{i^*i})(1 - \delta)$. For large $s$, $\boldsymbol{v}^{\odot(s-1)} \approx (0, 0, \dots, 1, 0, \dots, 0)$ with the 1 at the $i^*$-th position. Similarly, $|\boldsymbol{v}|_s^s \approx 1$. The $i^*$-th entry of $\Sigma \boldsymbol{v}$ is given by $\Sigma_{i^*i^*} - \sum_{j\neq i^*} |\Sigma_{i^*j}| + O(\delta)$. Hence for suitable sequences $\delta \to 0$ and $s \to \infty$, we obtain $\langle \boldsymbol{v}^{\odot(s-1)}, \Sigma \boldsymbol{v} \rangle / |\boldsymbol{v}|_s^s \to \Sigma_{i^*i^*} - \sum_{j\neq i^*} |\Sigma_{i^*j}| = \min_i \Sigma_{ii} - \sum_{j\neq i} |\Sigma_{ij}|$.

Regarding Assumption 3, we investigate the dependence of the Lipschitz constant $L_{s,q}$ on the dimension $d$ in high-dimensional linear regression models. If $s^*$ is the dual exponent of $s$, satisfying $1/s + 1/s^* = 1$, we show that the condition holds with

$$
L_{s,q} = \left\| |\boldsymbol{x}|_s |\boldsymbol{x}|_{s^*} \right\|_q. \tag{23}
$$

To see this, for any two vectors $\boldsymbol{\beta}, \boldsymbol{\beta}' \in \mathbb{R}^d$, we have

$$
\nabla g(\boldsymbol{\beta}, \boldsymbol{\xi}) - \nabla g(\boldsymbol{\beta}', \boldsymbol{\xi}) = -\left[(y - \boldsymbol{x}^\top \boldsymbol{\beta})\boldsymbol{x} - (y - \boldsymbol{x}^\top \boldsymbol{\beta}')\boldsymbol{x}\right] = \boldsymbol{x}\boldsymbol{x}^\top(\boldsymbol{\beta} - \boldsymbol{\beta}'). \tag{24}
$$

Taking the $\ell^s$-norm on both sides, we obtain

$$
\left|\nabla g(\boldsymbol{\beta}, \boldsymbol{\xi}) - \nabla g(\boldsymbol{\beta}', \boldsymbol{\xi})\right|_s = \left|\boldsymbol{x}\boldsymbol{x}^\top(\boldsymbol{\beta} - \boldsymbol{\beta}')\right|_s = |\boldsymbol{x}|_s \left|\boldsymbol{x}^\top(\boldsymbol{\beta} - \boldsymbol{\beta}')\right|. \tag{25}
$$

By Hölder's inequality, for the dual exponent $s^*$ satisfying $1/s + 1/s^* = 1$, it follows that

$$
\left|\boldsymbol{x}^\top(\boldsymbol{\beta} - \boldsymbol{\beta}')\right| \leq |\boldsymbol{x}|_{s^*} |\boldsymbol{\beta} - \boldsymbol{\beta}'|_s. \tag{26}
$$

Therefore, for $q \geq 2$, we have the $q$-th moment bounded as follows,

$$
\left(\mathbb{E}\left|\nabla g(\boldsymbol{\beta}, \boldsymbol{\xi}) - \nabla g(\boldsymbol{\beta}', \boldsymbol{\xi})\right|_s^q\right)^{1/q} \leq \left(\mathbb{E}\left[|\boldsymbol{x}|_s^q |\boldsymbol{x}|_{s^*}^q\right]\right)^{1/q} |\boldsymbol{\beta} - \boldsymbol{\beta}'|_s,
$$

proving (23).

Recall $s_d$ defined in (7). To bound the $\ell^\infty$-norm, we set the conjugates

$$
s = s_d, \quad s_d^* = \frac{s_d}{s_d - 1}. \tag{27}
$$

Recall that for the $\ell^s$-norm, we have $|\boldsymbol{x}|_\infty \leq |\boldsymbol{x}|_{s_d} \leq d^{1/s_d} |\boldsymbol{x}|_\infty \leq e|\boldsymbol{x}|_\infty$. Similarly, for the conjugate $\ell^{s_d^*}$-norm, $d^{\frac{1}{s_d^*}-1} = d^{\frac{1}{s_d}} \leq e$ implies

$$
\frac{1}{e}|\boldsymbol{x}|_1 \leq \frac{1}{d^{\frac{1}{s_d^*}-1}}|\boldsymbol{x}|_1 \leq |\boldsymbol{x}|_{s_d^*} \leq |\boldsymbol{x}|_1, \tag{28}
$$

which together with (23) gives

$$
L_{s_d, q} \leq e\left\| |\boldsymbol{x}|_\infty |\boldsymbol{x}|_1 \right\|_q. \tag{29}
$$

The next two lemmas show that the tail behavior of the covariate vector $\boldsymbol{x}_k$ determines the behavior of the Lipschitz constant $L_{s,q}$ and the moment $M_{s,q}$ defined in Assumption 3.

**Lemma 5.** *Consider the linear regression in* (18) *with i.i.d. generic random samples* $(\boldsymbol{x}, y)$, *where* $\boldsymbol{x} = (x_1, \ldots, x_d)^\top$. *Let* $q \geq 2$ *and recall* $s_d$ *in* (7).

(i) *(Sub-Gaussian) If there is a constant* $K$ *such that for all* $\boldsymbol{u} \in \mathbb{R}^d$, $|\boldsymbol{u}^\top \boldsymbol{x}|_{\psi_2} \leq K|\boldsymbol{u}|_2$, *where* $|v|_{\psi_2} = \inf\{t > 0 : \mathbb{E}[e^{v^2/t^2}] \leq 2\}$ *denotes the sub-Gaussian norm, then*

$$L_{s_d, q} = O(d\sqrt{\log(d)}).$$

(ii) *(Sub-exponential) If there is a constant* $K$ *such that for all* $\boldsymbol{u} \in \mathbb{R}^d$, $|\boldsymbol{u}^\top \boldsymbol{x}|_{\psi_1} \leq K|\boldsymbol{u}|_2$, *where* $|v|_{\psi_1} = \inf\{t > 0 : \mathbb{E}[e^{|v|/t}] \leq 2\}$ *denotes the sub-exponential norm, then*

$$L_{s_d, q} = O(d \log(d)).$$

(iii) *(Finite moment) If there is some* $p \geq 2q$ *and a finite constant* $K_p$ *such that for each* $1 \leq j \leq d$, $\mathbb{E}|x_j|^p \leq K_p$, *then*

$$L_{s_d, q} = O(d^{1 + \frac{1}{2q}}).$$

(iv) *For all three cases (i)–(iii), when* $s = 2$, $L_{2, q} = O(d)$.

*Proof of Lemma 5.* We write $\boldsymbol{x} = \boldsymbol{x}_k$ to denote a generic covariate. By (29) and Hölder's inequality,

$$L_{s_d, q} \leq e\big\||\boldsymbol{x}|_\infty|\boldsymbol{x}|_1\big\|_q \leq e\||\boldsymbol{x}|_\infty\|_{2q}\||\boldsymbol{x}|_1\|_{2q}.$$

The convexity of the function $t \mapsto t^{2q}$ and Jensen's inequality yield $|\boldsymbol{x}|_1^{2q} \leq d^{2q-1} \sum_{j=1}^d |x_j|^{2q}$ and

$$\mathbb{E}|\boldsymbol{x}|_1^{2q} \leq d^{2q-1} \sum_{j=1}^d \mathbb{E}|x_j|^{2q} \leq d^{2q} \max_{1 \leq j \leq d} \mathbb{E}|x_j|^{2q}.$$

Therefore, for all the three cases (i)–(iii),

$$\||\boldsymbol{x}|_1\|_{2q} = O(d).$$

Next, we study the order of $(\mathbb{E}[|\boldsymbol{x}|_\infty^{2q}])^{1/(2q)}$ for fixed $q \geq 2$.

(i) If each $x_j$ is sub-Gaussian, then by Section 2.5 in Vershynin [2018], we have

$$(\mathbb{E}[\max_{1 \leq j \leq d} |x_j|^{2q}])^{1/(2q)} \leq K(\sqrt{\log(d)} + \sqrt{q}) = O(\sqrt{\log(d)}).$$

(ii) If each $x_j$ is sub-exponential, then by Section 2.7 in Vershynin [2018], we obtain

$$(\mathbb{E}[\max_{1 \leq j \leq d} |x_j|^{2q}])^{1/(2q)} = O(K(\log(d) + \log(q))) = O(\log(d)).$$

(iii) If each $x_j$ has the finite $p$-th moment for some $p \geq 2q$, then

$$\mathbb{E}[\max_{1 \leq j \leq d} |x_j|^{2q}] \leq \sum_{1 \leq j \leq d} \mathbb{E}[|x_j|^{2q}] \leq dK_q = O(d).$$

Finally, for case (iv) with $s_d = 2$, by (23),

$$L_{2, q} = \||\boldsymbol{x}|_2\|_{2q}^2.$$

By the convexity of the function $t \mapsto t^q$, we apply Jensen's inequality and obtain

$$|\boldsymbol{x}|_2^{2q} = \Big(\sum_{j=1}^d x_j^2\Big)^q \leq d^{q-1} \sum_{j=1}^d |x_j|^{2q}.$$

Therefore, for $\boldsymbol{x}$ satisfying case (iii),

$$\mathbb{E}|\boldsymbol{x}|_2^{2q} \leq d^{q-1} \sum_{j=1}^d \mathbb{E}|x_j|^{2q} \leq d^q K_p, \tag{30}$$

which yields $L_{2,q} = O(d)$. For the cases (i) and (ii), by Sections 3.4 and 2.7 in Vershynin [2018], respectively, we obtain

$$\||\boldsymbol{x}|_2\|_{2q} = O(K(\sqrt{d} + \sqrt{q})) = O(\sqrt{d}),$$

and

$$\||\boldsymbol{x}|_2\|_{2q} = O(K(q\sqrt{d})) = O(\sqrt{d}),$$

both indicating $L_{2,q} = O(d)$. This completes the proof. $\qquad\square$

**Lemma 6.** *Consider the linear regression model in* (18) *and assume the conditions on $\epsilon$ and $\boldsymbol{x}$ therein are satisfied. Recall that $M_{s,q} = \||\nabla g(\boldsymbol{\beta}^*, \boldsymbol{\xi})|_s\|_q$ is defined in Assumption 3 for some $q \geq 2$. For the same four cases (i)–(iv) as in Lemma 5 and $s_d$ defined in* (7), *$M_{s_d,q}$ is respectively equal to* (i) $O(\sqrt{\log(d)})$, (ii) $O(\log(d))$, (iii) $O(d^{1/(2q)})$ *and* (iv) $O(\sqrt{d})$.

*Proof of Lemma 6.* In the linear regression model, the stochastic gradient at the global minimum $\boldsymbol{\beta}^*$ can be rewritten into

$$\nabla g(\boldsymbol{\beta}^*, \boldsymbol{\xi}) = -(y - \boldsymbol{x}^\top \boldsymbol{\beta}^*) = -\epsilon \boldsymbol{x}.$$

Since the noise $\epsilon$ is independent of the covariate vector $\boldsymbol{x}$, we obtain

$$\||\nabla g(\boldsymbol{\beta}^*, \boldsymbol{\xi})|_{s_d}\|_q = \||\epsilon| \cdot |\boldsymbol{x}|_{s_d}\|_q = \|\epsilon\|_q \||\boldsymbol{x}|_{s_d}\|_q.$$

By inequality (8), it suffices to bound $\||\boldsymbol{x}|_\infty\|_q$. Since $\||\boldsymbol{x}|_\infty\|_q \leq \||\boldsymbol{x}|_\infty\|_{2q}$, the same arguments in the proof of Lemma 5 carry over immediately. We omit the details here. $\qquad\square$

## 8.3 Some Useful Lemmas

**Lemma 7** (Maximal inequality [Chernozhukov et al., 2015]). *Let $\boldsymbol{z}_1, \ldots, \boldsymbol{z}_n$ be independent, $d$-dimensional random vectors. Denote the $j$-th element of $\boldsymbol{z}_i$ by $z_{ij}$, $1 \leq j \leq d$. Define $M := \max_{1 \leq i \leq n} \max_{1 \leq j \leq d} |z_{ij}|$ and $\sigma^2 := \max_{1 \leq j \leq d} \sum_{i=1}^n \mathbb{E}[z_{ij}^2]$. Then,*

$$\mathbb{E}\big[\max_{1 \leq j \leq d} |\sum_{i=1}^n (z_{ij} - \mathbb{E}[z_{ij}])|\big] \lesssim \sigma\sqrt{\log(d)} + \sqrt{\mathbb{E}[M^2]}\log(d),$$

*where the universal constant in $\lesssim$ is positive and independent of $n$ and $d$.*

**Lemma 8** ($L^q$ maximal inequality). *Let $\boldsymbol{x}_1, \ldots, \boldsymbol{x}_n$ be independent, $d$-dimensional random vectors. Denote by $x_{ij}$ the $j$-th element of $\boldsymbol{x}_i$, $1 \leq j \leq d$. Then,*

$$\Big\|\max_{1 \leq j \leq d} \big|\sum_{i=1}^n (x_{ij} - \mathbb{E}[x_{ij}])\big|\Big\|_q^2 \leq e^2\big(\max\{q, \log(d)\} - 1\big) \sum_{i=1}^n \Big\|\max_{1 \leq j \leq d} |x_{ij} - \mathbb{E}[x_{ij}]|\Big\|_q^2.$$

This moment inequality can be derived by repeatedly applying Lemma 2. It generalizes the maximal inequality for $\mathbb{E}[\max_{1 \leq j \leq d} |\sum_{i=1}^n (x_{ij} - \mathbb{E}[x_{ij}])|]$ in Chernozhukov et al. [2015], reproduced above as Lemma 7, to general $q$-th moments.

*Proof of Lemma 8.* One can assume that the independent random vectors $\boldsymbol{x}_1, \ldots, \boldsymbol{x}_n$ have zero means. By repeatedly applying Lemma 2 and choosing $s = \log(d)$,

$$
\begin{aligned}
\big\||\boldsymbol{x}_1 + \cdots + \boldsymbol{x}_n|_\infty\big\|_q^2 &\leq \big\||\boldsymbol{x}_1 + \cdots + \boldsymbol{x}_n|_s\big\|_q^2 \\
&\leq \big\||\boldsymbol{x}_1 + \cdots + \boldsymbol{x}_{n-1}|_s\big\|_q^2 + (\max\{q, s\} - 1)\big\||\boldsymbol{x}_n|_s\big\|_q^2 \\
&\leq (\max\{q, s\} - 1) \sum_{i=1}^n \big\||\boldsymbol{x}_i|_s\big\|_q^2 \\
&\leq e^2\big(\max\{q, \log(d)\} - 1\big) \sum_{i=1}^n \big\||\boldsymbol{x}_i|_\infty\big\|_q^2.
\end{aligned}
$$

$\qquad\square$

**Lemma 9** (Moment inequality [Li et al., 2024c]). *Let $q \geq 2$. For any two random vectors $\boldsymbol{x}$ and $\boldsymbol{y}$ in $\mathbb{R}^d$ with fixed $d \geq 1$, and let*

$$\Delta = \mathbb{E}\Big| \|\boldsymbol{x} + \boldsymbol{y}\|_2^q - \|\boldsymbol{x}\|_2^q - q\|\boldsymbol{x}\|_2^{q-2}\boldsymbol{x}^\top\boldsymbol{y} \Big|.$$

*Then, the following inequalities holds:*

*(i)*

$$\Delta \leq \mathbb{E}\big(\|\boldsymbol{x}\|_2 + \|\boldsymbol{y}\|_2\big)^q - \mathbb{E}\|\boldsymbol{x}\|_2^q - q\mathbb{E}(\|\boldsymbol{x}\|_2^{q-1}\|\boldsymbol{y}\|_2).$$

*(ii)*

$$\Delta \leq \big[(\mathbb{E}\|\boldsymbol{x}\|_2^q)^{1/q} + (\mathbb{E}\|\boldsymbol{y}\|_2^q)^{1/q}\big]^q - \mathbb{E}\|\boldsymbol{x}\|_2^q - q(\mathbb{E}\|\boldsymbol{x}\|_2^q)^{(q-1)/q}(\mathbb{E}\|\boldsymbol{y}\|_2^q)^{1/q}.$$

**Lemma 10** (Equivalence of $\ell^s$-$\ell^\infty$-induced matrix norms). *For matrix $A \in \mathbb{R}^{d \times d}$, we have the equivalence of the $\ell^{s_d}$-norm and $\ell^\infty$-norm induced matrix norms as follows*

$$\frac{1}{e}\|A\|_\infty \leq \|A\|_{s_d} \leq e\|A\|_\infty, \tag{31}$$

*where $s_d$ is defined as (7) and $\|A\|_s = \max_{|\boldsymbol{x}|_s \neq 0} |A\boldsymbol{x}|_s / |\boldsymbol{x}|_s$. If in addition, $A$ is symmetric, then*

$$\frac{1}{e}\|A\|_1 = \frac{1}{e}\|A\|_\infty \leq \|A\|_{s_d} \leq e\|A\|_\infty = e\|A\|_1. \tag{32}$$

*Proof of Lemma 10.* By Horn and Johnson [1985], for any $1 \leq p \leq q \leq \infty$ and matrix $A \in \mathbb{R}^{d \times d}$,

$$d^{(1/q)-(1/p)}\|A\|_q \leq \|A\|_p \leq d^{(1/p)-(1/q)}\|A\|_q. \tag{33}$$

For $p = s$ and $q = \infty$, we obtain

$$d^{-1/s}\|A\|_\infty \leq \|A\|_s \leq d^{1/s}\|A\|_\infty. \tag{34}$$

Since $d^{1/s} \leq e$ by choosing $s = s_d$ in (7), we obtain (31).

For symmetric $A = (a_{ij})_{1 \leq i,j \leq d}$, $a_{ij} = a_{ji}$ for all $i, j$. Therefore,

$$\|A\|_1 = \max_{1 \leq j \leq d} \sum_{i=1}^d |a_{ij}| = \max_{1 \leq i \leq d} \sum_{j=1}^d |a_{ij}| = \|A\|_\infty. \tag{35}$$

This completes the proof. $\qquad\square$

## 8.4 Proofs for Section 2

*Derivation of* (15)*:* Since $s$ is an even integer, we can write

$$f(\alpha) := |\boldsymbol{x} - \alpha\boldsymbol{z}|_s^q = \Big\{\sum_{i=1}^d (x_i - \alpha z_i)^s\Big\}^{\frac{q}{s}}. \tag{36}$$

Taking the derivative with respect to $\alpha$, we obtain

$$\begin{aligned}
f'(\alpha) := \frac{d}{d\alpha}f(\alpha) &= \frac{q}{s}\Big\{\sum_{i=1}^d (x_i - \alpha z_i)^s\Big\}^{\frac{q}{s}-1}\sum_{i=1}^d \frac{d}{d\alpha}(x_i - \alpha z_i)^s \\
&= \frac{q}{s}\Big\{\sum_{i=1}^d (x_i - \alpha z_i)^s\Big\}^{\frac{q}{s}-1}\sum_{i=1}^d s(x_i - \alpha z_i)^{s-1}(-z_i) \\
&= -q\Big\{\sum_{i=1}^d (x_i - \alpha z_i)^s\Big\}^{\frac{q}{s}-1}\sum_{i=1}^d (x_i - \alpha z_i)^{s-1} z_i. \tag{37}
\end{aligned}$$

Therefore,

$$\begin{aligned}
f'(0) &= -q\Big\{\sum_{i=1}^d x_i^s\Big\}^{\frac{q}{s}-1}\sum_{i=1}^d x_i^{s-1} z_i \\
&= -q|\boldsymbol{x}|_s^{q-s}\sum_{i=1}^d x_i^{s-1} z_i. \tag{38}
\end{aligned}$$

A first-order Taylor expansion yields then (15).

*Proof of Lemma 1.* Recall that we have defined

$$f(\alpha) = |\boldsymbol{x} - \alpha \boldsymbol{z}|_s^q = \Big\{ \sum_{i=1}^d (x_i - \alpha z_i)^s \Big\}^{\frac{q}{s}}.$$

A second order Taylor expansion gives $f(\alpha) = f(0) + \alpha f'(0) + \frac{1}{2}\alpha^2 f''(\eta)$ for some $\eta \in [0, \alpha]$. It suffices to bound $\sup_{u \in [0,\alpha]} |f''(u)|$. Defining

$$M(u) := \sum_{i=1}^d (x_i - u z_i)^s = |\boldsymbol{x} - u\boldsymbol{z}|_s^s, \tag{39}$$

we have $f(u) = [M(u)]^{\frac{q}{s}}$,

$$M'(u) = -s \sum_{i=1}^d (x_i - u z_i)^{s-1} z_i, \tag{40}$$

$$M''(u) = s(s-1) \sum_{i=1}^d (x_i - u z_i)^{s-2} z_i^2, \tag{41}$$

and the first two derivatives of $f(u)$ can be respectively expressed by

$$f'(u) = \frac{q}{s}[M(u)]^{\frac{q}{s}-1} M'(u), \tag{42}$$

$$f''(u) = \frac{q}{s}\Big(\frac{q}{s}-1\Big)[M(u)]^{\frac{q}{s}-2}[M'(u)]^2 + M''(u)\frac{q}{s}[M(u)]^{\frac{q}{s}-1}. \tag{43}$$

Since $s$ is an even integer, it follows from Hölder's inequality that

$$[M'(u)]^2 = s^2 \Big( \sum_{i=1}^d (x_i - u z_i)^{s-1} z_i \Big)^2$$

$$\leq s^2 \Big( \Big( \sum_{i=1}^d (x_i - u z_i)^s \Big)^{\frac{s-1}{s}} \Big( \sum_{i=1}^d z_i^s \Big)^{1/s} \Big)^2$$

$$= s^2 |\boldsymbol{x} - u\boldsymbol{z}|_s^{2(s-1)} |\boldsymbol{z}|_s^2. \tag{44}$$

By applying Hölder's inequality again, we obtain

$$\big|M''(u)\big| \leq s(s-1)\Big( \sum_{i=1}^d (x_i - u z_i)^s \Big)^{\frac{s-2}{s}} \Big( \sum_{i=1}^d z_i^s \Big)^{2/s}$$

$$= s(s-1)|\boldsymbol{x} - u\boldsymbol{z}|_s^{s-2}|\boldsymbol{z}|_s^2. \tag{45}$$

By the two results above, we have

$$|f''(u)| = \Big| \frac{q}{s}\Big(\frac{q}{s}-1\Big)|\boldsymbol{x} - u\boldsymbol{z}|_s^{q-2s}[M'(u)]^2 + M''(u)\frac{q}{s}|\boldsymbol{x} - u\boldsymbol{z}|_s^{q-s} \Big|$$

$$\leq q|q-s| \cdot |\boldsymbol{x} - u\boldsymbol{z}|_s^{q-2}|\boldsymbol{z}|_s^2 + q(s-1)|\boldsymbol{x} - u\boldsymbol{z}|_s^{q-2}|\boldsymbol{z}|_s^2$$

$$\leq q\big[|q-s| + (s-1)\big]\big(|\boldsymbol{x}|_s + |u\boldsymbol{z}|_s\big)^{q-2}|\boldsymbol{z}|_s^2. \tag{46}$$

Since $u \in [0, \alpha]$, it follows that

$$\sup_{u \in [0,\alpha]} |f''(u)| \leq q\big[|q-s| + (s-1)\big]\big(|\boldsymbol{x}|_s + \alpha|\boldsymbol{z}|_s\big)^{q-2}|\boldsymbol{z}|_s^2. \tag{47}$$

This completes the proof. $\qquad\square$

*Existence of solution to* (13)*:* To see the existence of the solution $\alpha_{s,q}$ in

$$1 - q\mu\alpha + \frac{q\big[|q-s| + (s-1)\big]L_{s,q}^2}{2}\alpha^2(1 + \alpha L_{s,q})^{q-2} = 1.$$

denote the function $\alpha \mapsto F(\alpha) = -\mu + c\alpha(1+L)^{q-2}$ for the constant $c = [|q-s| + (s-1)]L^2/2 > 0$ and $L = L_{s,q}$. For any $q \geq 2$, and any $\alpha > 0$, $F'(\alpha) = c[(1+L\alpha)^{q-2} + \alpha(q-2)L(1+L\alpha)^{q-3}] > 0$, proving that $F(\alpha)$ is strictly increasing on $\alpha > 0$. Since $F(0) = -\mu < 0$ and $F(\infty) = +\infty$, the unique root to $F(\alpha) = 0$ exists.

*Proof of Lemma 2.* Define $\varphi(t) = \|\,|\boldsymbol{x} + t\boldsymbol{y}|_s\|_q^2$ for $t \in [0,1]$. Then

$$\varphi'(t) = \frac{2}{q}\Big[\mathbb{E}\Big\{\sum_{j=1}^{d}(x_j + ty_j)^s\Big\}^{q/s}\Big]^{2/q-1}\frac{q}{s}\mathbb{E}\Big[\Big\{\sum_{j=1}^{d}(x_j + ty_j)^s\Big\}^{q/s-1}\sum_{j=1}^{d}s(x_j + ty_j)^{s-1}y_j\Big]$$

$$= 2\Big[\mathbb{E}\Big\{\sum_{j=1}^{d}(x_j + ty_j)^s\Big\}^{q/s}\Big]^{2/q-1}\mathbb{E}\Big[\Big\{\sum_{j=1}^{d}(x_j + ty_j)^s\Big\}^{q/s-1}\sum_{j=1}^{d}(x_j + ty_j)^{s-1}y_j\Big]$$

and

$$\varphi''(t) = 2\Big(\frac{2}{q} - 1\Big)\Big[\mathbb{E}\Big\{\sum_{j=1}^{d}(x_j + ty_j)^s\Big\}^{q/s}\Big]^{2/q-2}$$

$$\cdot \frac{q}{s}\cdot s\Big|\mathbb{E}\Big[\Big\{\sum_{j=1}^{d}(x_j + ty_j)^s\Big\}^{q/s-1}\sum_{j=1}^{d}(x_j + ty_j)^{s-1}y_j\Big]\Big|^2$$

$$+ 2\Big[\mathbb{E}\Big\{\sum_{j=1}^{d}(x_j + ty_j)^s\Big\}^{q/s}\Big]^{2/q-1}$$

$$\cdot \mathbb{E}\Big[\Big(\frac{q}{s} - 1\Big)\Big\{\sum_{j=1}^{d}(x_j + ty_j)^s\Big\}^{q/s-2}s\Big\{\sum_{j=1}^{d}(x_j + ty_j)^{s-1}y_j\Big\}^2\Big]$$

$$+ 2\Big[\mathbb{E}\Big\{\sum_{j=1}^{d}(x_j + ty_j)^s\Big\}^{q/s}\Big]^{2/q-1}\mathbb{E}\Big[\Big\{\sum_{j=1}^{d}(x_j + ty_j)^s\Big\}^{q/s-1}(s-1)\sum_{j=1}^{d}(x_j + ty_j)^{s-2}y_j^2\Big]$$

$$=: \Delta_1(t) + \Delta_2(t) + \Delta_3(t)$$

Since $q \geq 2$, $\Delta_1(t) \leq 0$.

**Case I.** If $q/s - 1 \leq 0$, then $\Delta_2(t) \leq 0$ and $\varphi''(t) \leq \Delta_3(t)$. By Hölder's inequality,

$$\sum_{j=1}^{d}(x_j + ty_j)^{s-2}y_j^2 \leq \Big\{\sum_{j=1}^{d}(x_j + ty_j)^s\Big\}^{(s-2)/s}\Big(\sum_{j=1}^{d}y_j^s\Big)^{2/s}$$

Consequently,

$$\Delta_3(t) \leq 2(s-1)\Big[\mathbb{E}\Big\{\sum_{j=1}^{d}(x_j + ty_j)^s\Big\}^{q/s}\Big]^{2/q-1}$$

$$\cdot \mathbb{E}\Big[\Big\{\sum_{j=1}^{d}(x_j + ty_j)^s\Big\}^{q/s-1}\Big\{\sum_{j=1}^{d}(x_j + ty_j)^s\Big\}^{(s-2)/s}\Big(\sum_{j=1}^{d}y_j^s\Big)^{2/s}\Big]$$

$$= 2(s-1)\Big[\mathbb{E}\Big\{\sum_{j=1}^{d}(x_j + ty_j)^s\Big\}^{q/s}\Big]^{2/q-1}\mathbb{E}\Big[\Big\{\sum_{j=1}^{d}(x_j + ty_j)^s\Big\}^{(q-2)/s}\Big(\sum_{j=1}^{d}y_j^s\Big)^{2/s}\Big]$$

$$= 2(s-1)\|\,|\boldsymbol{x} + t\boldsymbol{y}|_s\|_q^{2-q}\mathbb{E}\Big[\Big\{\sum_{j=1}^{d}(x_j + ty_j)^s\Big\}^{(q-2)/s}\Big(\sum_{j=1}^{d}y_j^s\Big)^{2/s}\Big]$$

$$\leq 2(s-1)\|\,|\boldsymbol{x} + t\boldsymbol{y}|_s\|_q^{2-q}\|\,|\boldsymbol{x} + t\boldsymbol{y}|_s\|_q^{q-2}\|\,|\boldsymbol{y}|_s\|_q^2$$

$$= 2(s-1)\|\,|\boldsymbol{y}|_s\|_q^2.$$

**Case II.** If $q/s - 1 > 0$, by Hölder's inequality,

$$\Big\{\sum_{j=1}^{d}(x_j + ty_j)^{s-1}y_j\Big\}^2 = \Big\{\sum_{j=1}^{d}(x_j + ty_j)^{s/2}(x_j + ty_j)^{s/2-1}y_j\Big\}^2$$

$$\leq \sum_{j=1}^{d}(x_j + ty_j)^s\sum_{j=1}^{d}(x_j + ty_j)^{s-2}y_j^2.$$

Therefore,

$$\Delta_2(t) \le \Delta_3(t) \frac{q-s}{s-1}$$

and

$$\varphi''(t) \le \Delta_2(t) + \Delta_3(t) \le \Delta_3(t) \frac{q-1}{s-1} \le 2(q-1)\||\boldsymbol{y}|_s\|_q^2.$$

Then, we have

$$\||\boldsymbol{x}+\boldsymbol{y}|_s\|_q^2 = \varphi(1) = \varphi(0) + \varphi'(0) + \int_0^1 (1-t)\varphi''(t)\,dt$$

$$\le \||\boldsymbol{x}|_s\|_q^2 + 2\||\boldsymbol{x}|_s\|_q^{2-q}\mathbb{E}\Big(|\boldsymbol{x}|_s^{q-s}\sum_{j=1}^d x_j^{s-1}y_j\Big) + \big(\max\{q,s\}-1\big)\||\boldsymbol{y}|_s\|_q^2.$$

$\square$

*Proof of Theorem 1.* Consider the iterated random function

$$F : \mathbb{R}^d \times \mathbb{R} \mapsto \mathbb{R}, \quad (\boldsymbol{\beta}, \boldsymbol{\xi}) \mapsto F_{\boldsymbol{\xi}}(\boldsymbol{\beta}) = \boldsymbol{\beta} - \alpha\nabla g(\boldsymbol{\beta}, \boldsymbol{\xi}). \tag{48}$$

To prove GMC in Theorem 1, it suffices to show that, for some $q \ge 2$ and even integer $s \ge 2$, for any fixed vectors $\boldsymbol{\beta}, \boldsymbol{\beta}' \in \mathbb{R}^d$,

$$\||F_{\boldsymbol{\xi}}(\boldsymbol{\beta}) - F_{\boldsymbol{\xi}}(\boldsymbol{\beta}')|_s\|_q \le r_{\alpha,s,q}|\boldsymbol{\beta}-\boldsymbol{\beta}'|_s.$$

Recall the inequality in Lemma 2. For $\boldsymbol{x}$ and $\boldsymbol{y}$ therein, we choose them to be $\boldsymbol{x} = \boldsymbol{\beta}-\boldsymbol{\beta}'$ and $\boldsymbol{y} = -\alpha(\nabla g(\boldsymbol{\beta}, \boldsymbol{\xi}) - \nabla g(\boldsymbol{\beta}', \boldsymbol{\xi}))$ respectively. Then, it directly follows from Lemma 2 that

$$\||F_{\boldsymbol{\xi}}(\boldsymbol{\beta}) - F_{\boldsymbol{\xi}}(\boldsymbol{\beta}')|_s\|_q^2$$
$$\le |\boldsymbol{\beta}-\boldsymbol{\beta}'|_s^2 - 2\alpha|\boldsymbol{\beta}-\boldsymbol{\beta}'|_s^{2-q}\mathbb{E}\Big[|\boldsymbol{\beta}-\boldsymbol{\beta}'|_s^{q-s}\big\langle(\boldsymbol{\beta}-\boldsymbol{\beta}')^{s-1}, \nabla g(\boldsymbol{\beta}, \boldsymbol{\xi}) - \nabla g(\boldsymbol{\beta}', \boldsymbol{\xi})\big\rangle\Big]$$
$$\quad + \alpha^2(\max\{q,s\}-1)\||\nabla g(\boldsymbol{\beta}, \boldsymbol{\xi}) - \nabla g(\boldsymbol{\beta}', \boldsymbol{\xi})|_s\|_q^2$$
$$= |\boldsymbol{\beta}-\boldsymbol{\beta}'|_s^2 - 2\alpha|\boldsymbol{\beta}-\boldsymbol{\beta}'|_s^{2-s}\big\langle(\boldsymbol{\beta}-\boldsymbol{\beta}')^{s-1}, G(\boldsymbol{\beta}) - G(\boldsymbol{\beta}')\big\rangle$$
$$\quad + \alpha^2(\max\{q,s\}-1)\||\nabla g(\boldsymbol{\beta}, \boldsymbol{\xi}) - \nabla g(\boldsymbol{\beta}', \boldsymbol{\xi})|_s\|_q^2.$$

This along with Assumptions 2 and 3 yields

$$\||F_{\boldsymbol{\xi}}(\boldsymbol{\beta}) - F_{\boldsymbol{\xi}}(\boldsymbol{\beta}')|_s\|_q^2 \le \big(1 - 2\alpha\mu + \alpha^2(\max\{q,s\}-1)L_{s,q}^2\big)|\boldsymbol{\beta}-\boldsymbol{\beta}'|_s^2,$$

which completes the proof. $\square$

## 8.5 Proofs for Section 3.1

*Proof of Proposition 1.* Recall (17) and let $\nabla G(\boldsymbol{\beta}) = \big(\nabla G_1(\boldsymbol{\beta}), \ldots, \nabla G_d(\boldsymbol{\beta})\big)^\top$ with

$$\nabla G_i(\boldsymbol{\beta}) = \partial G(\boldsymbol{\beta})/\partial\beta_i = \big(\mathbb{E}[\nabla g(\boldsymbol{\beta}, \boldsymbol{\xi})]\big)_i, \quad i = 1, \ldots, d. \tag{49}$$

Since the random samples $\boldsymbol{\xi}_k$, $k \ge 1$, are independent, it follows that for the $k$-th iteration, $\boldsymbol{\xi}_k$ is independent of $\boldsymbol{\beta}_{k-1}$. Then, by the tower rule, for all $k \ge 1$,

$$\mathbb{E}_{\boldsymbol{\xi}}\big[\nabla g(\boldsymbol{\beta}_{k-1}, \boldsymbol{\xi}_k) - \nabla G(\boldsymbol{\beta}_{k-1}) \mid \boldsymbol{\beta}_{k-1}\big] = \mathbb{E}_{\boldsymbol{\xi}}[\nabla g(\boldsymbol{\beta}_{k-1}, \boldsymbol{\xi}_k) - \nabla G(\boldsymbol{\beta}_{k-1})] = 0. \tag{50}$$

Therefore, by applying the high-dimensional moment inequality (16) in Lemma 2, we obtain

$$\||\boldsymbol{\beta}_k - \boldsymbol{\beta}^*|_s\|_q^2 \le \||\boldsymbol{\beta}_{k-1} - \boldsymbol{\beta}^* - \alpha\nabla G(\boldsymbol{\beta}_{k-1})|_s\|_q^2$$
$$\quad + (\max\{q,s\}-1)\alpha^2\||\nabla g(\boldsymbol{\beta}_{k-1}, \boldsymbol{\xi}_k) - \nabla G(\boldsymbol{\beta}_{k-1})|_s\|_q^2. \tag{51}$$

For the second part in (51), noting that $\nabla G(\boldsymbol{\beta}^*) = 0$, by the triangle inequality, we have

$$\left\|\left|\nabla g(\boldsymbol{\beta}_{k-1}, \boldsymbol{\xi}_k) - \nabla G(\boldsymbol{\beta}_{k-1})\right|_s\right\|_q^2$$

$$\leq \left(\left\|\left|\nabla g(\boldsymbol{\beta}_{k-1}, \boldsymbol{\xi}_k) - \nabla g(\boldsymbol{\beta}^*, \boldsymbol{\xi}_k)\right|_s\right\|_q + \left\|\left|\nabla G(\boldsymbol{\beta}_{k-1}) - \nabla G(\boldsymbol{\beta}^*)\right|_s\right\|_q + \left\|\left|\nabla g(\boldsymbol{\beta}^*, \boldsymbol{\xi}_k)\right|_s\right\|_q\right)^2$$

$$\leq 3\left\|\left|\nabla g(\boldsymbol{\beta}_{k-1}, \boldsymbol{\xi}_k) - \nabla g(\boldsymbol{\beta}^*, \boldsymbol{\xi}_k)\right|_s\right\|_q^2 + 3\left\|\left|\nabla G(\boldsymbol{\beta}_{k-1}) - \nabla G(\boldsymbol{\beta}^*)\right|_s\right\|_q^2 + 3\left\|\left|\nabla g(\boldsymbol{\beta}^*, \boldsymbol{\xi}_k)\right|_s\right\|_q^2.$$
(52)

Since $|\cdot|_s$ is a convex function for $s \geq 1$, we have $|\mathbb{E}[\cdot]|_s \leq \mathbb{E}[|\cdot|_s]$. Thus, for all $q \geq 1$, by Jensen's inequality, we can bound

$$|\nabla G(\boldsymbol{\beta}_{k-1}) - \nabla G(\boldsymbol{\beta}^*)|_s = \left|\mathbb{E}_{\boldsymbol{\xi}}\left[\nabla g(\boldsymbol{\beta}_{k-1}, \boldsymbol{\xi}_k) - \nabla g(\boldsymbol{\beta}^*, \boldsymbol{\xi}_k)\right]\right|_s$$

$$\leq \mathbb{E}_{\boldsymbol{\xi}}\left[\left|\nabla g(\boldsymbol{\beta}_{k-1}, \boldsymbol{\xi}_k) - \nabla g(\boldsymbol{\beta}^*, \boldsymbol{\xi}_k)\right|_s\right]$$

$$\leq \left(\mathbb{E}_{\boldsymbol{\xi}}\left|\nabla g(\boldsymbol{\beta}_{k-1}, \boldsymbol{\xi}_k) - \nabla g(\boldsymbol{\beta}^*, \boldsymbol{\xi}_k)\right|_s^q\right)^{1/q}.$$
(53)

This along with Assumption 3 yields

$$\left\|\left|\nabla G(\boldsymbol{\beta}_{k-1}) - \nabla G(\boldsymbol{\beta}^*)\right|_s\right\|_q \leq \left(\mathbb{E}_{\boldsymbol{\beta}}\mathbb{E}_{\boldsymbol{\xi}}\left|\nabla g(\boldsymbol{\beta}_{k-1}, \boldsymbol{\xi}_k) - \nabla g(\boldsymbol{\beta}^*, \boldsymbol{\xi}_k)\right|_s^q\right)^{1/q}$$

$$= \left\|\left|\nabla g(\boldsymbol{\beta}_{k-1}, \boldsymbol{\xi}_k) - \nabla g(\boldsymbol{\beta}^*, \boldsymbol{\xi}_k)\right|_s\right\|_q$$

$$\leq L_{s,q}\left\|\left|\boldsymbol{\beta}_{k-1} - \boldsymbol{\beta}^*\right|_s\right\|_q.$$
(54)

Inserting this result back into (52), we obtain a bound for the second term in (51) using

$$\left\|\left|\nabla g(\boldsymbol{\beta}_{k-1}, \boldsymbol{\xi}_k) - \nabla G(\boldsymbol{\beta}_{k-1})\right|_s\right\|_q^2 \leq 6L_{s,q}^2\left\|\left|\boldsymbol{\beta}_{k-1} - \boldsymbol{\beta}^*\right|_s\right\|_q^2 + 3\left\|\left|\nabla g(\boldsymbol{\beta}^*, \boldsymbol{\xi}_k)\right|_s\right\|_q^2.$$
(55)

For the first term in (51), by applying Lemma 2 again, it follows from Assumptions 2 and 3 that

$$\left\|\left|\boldsymbol{\beta}_{k-1} - \boldsymbol{\beta}^* - \alpha\nabla G(\boldsymbol{\beta}_{k-1})\right|_s\right\|_q^2$$

$$\leq \left\|\left|\boldsymbol{\beta}_{k-1} - \boldsymbol{\beta}^*\right|_s\right\|_q^2 - 2\alpha\left\|\left|\boldsymbol{\beta}_{k-1} - \boldsymbol{\beta}^*\right|_s\right\|_q^{2-q}\mathbb{E}\left(\left|\boldsymbol{\beta}_{k-1} - \boldsymbol{\beta}^*\right|_s^{q-s}\sum_{j=1}^d(\boldsymbol{\beta}_{k-1} - \boldsymbol{\beta}^*)_j^{s-1}\nabla G_j(\boldsymbol{\beta}_{k-1})\right)$$

$$+ \alpha^2(\max\{q, s\} - 1)\left\|\left|\nabla G(\boldsymbol{\beta}_{k-1}) - \nabla G(\boldsymbol{\beta}^*)\right|_s\right\|_q^2$$

$$\leq \left(1 - 2\alpha\mu + \alpha^2(\max\{q, s\} - 1)L_{s,q}^2\right)\left\|\left|\boldsymbol{\beta}_{k-1} - \boldsymbol{\beta}^*\right|_s\right\|_q^2.$$
(56)

Inserting this inequality and (55) into (51), we obtain the inequality

$$\|\left|\boldsymbol{\beta}_k - \boldsymbol{\beta}^*\right|_s\|_q^2 \leq \left(1 - 2\alpha\mu + 7(\max\{q, s\} - 1)\alpha^2 L_{s,q}^2\right)\|\left|\boldsymbol{\beta}_{k-1} - \boldsymbol{\beta}^*\right|_s\|_q^2$$

$$+ 3(\max\{q, s\} - 1)\alpha^2\|\left|\nabla g(\boldsymbol{\beta}^*, \boldsymbol{\xi}_k)\right|_s\|_q^2.$$

The desired result is achieved since $\|\left|\nabla g(\boldsymbol{\beta}^*, \boldsymbol{\xi}_k)\right|_s\|_q \leq M_{s,q}$ by Assumption 3. As a special case, for the stationary SGD iterates $\boldsymbol{\beta}_k^\circ \sim \pi_\alpha$, $k \geq 1$, we obtain the same result. $\qquad\square$

*Proof of Theorem 2.* First, we denote the contraction constant in Proposition 1 as follows

$$\tilde{r}_{\alpha,s,q} := 1 - 2\alpha\mu + 7(\max\{q, s\} - 1)\alpha^2 L_{s,q}^2.$$
(57)

Given the range of the constant learning rate $\alpha$, we have $\tilde{r}_{\alpha,s,q} < 1$. Moreover, notice that

$$3(\max\{q, s\} - 1)\alpha^2\|\left|\nabla g(\boldsymbol{\beta}^*, \boldsymbol{\xi}_k)\right|_s\|_q^2 = O\left(\max\{q, s\}\alpha^2 M_{s,q}^2\right).$$
(58)

Therefore, for the stationary SGD iterates $\boldsymbol{\beta}_k^\circ \sim \pi_\alpha$, by Proposition 1, we can obtain

$$\|\left|\boldsymbol{\beta}_k^\circ - \boldsymbol{\beta}^*\right|_s\|_q^2 \leq \tilde{r}_{\alpha,s,q}\|\left|\boldsymbol{\beta}_{k-1}^\circ - \boldsymbol{\beta}^*\right|_s\|_q^2 + O\left(\max\{q, s\}\alpha^2 M_{s,q}^2\right).$$
(59)

Since the SGD iterates $\beta_k^\circ$ satisfy the geometric-moment contraction in Theorem 1, following Remark 2 in Wu and Shao [2004], the recursion $\beta_k^\circ = \beta_{k-1}^\circ - \alpha \nabla g(\beta_{k-1}^\circ, \xi_k)$ also holds for $k \leq 0$. Thus, we can recursively apply the inequality above and achieve

$$\||\beta_k^\circ - \beta^*|_s\|_q^2 \leq O\big(\max\{q, s\}\alpha^2 M_{s,q}^2\big) \cdot \sum_{i=0}^\infty \tilde{r}_{\alpha,s,q}^i$$

$$= \frac{1}{1 - \tilde{r}_{\alpha,s,q}} O\big(\max\{q, s\}\alpha^2 M_{s,q}^2\big)$$

$$= O\big(\max\{q, s\}\alpha M_{s,q}^2\big). \tag{60}$$

This finishes the proof for the stationary SGD sequence.

Furthermore, for the general SGD iterates $\beta_k$ in (2) that may not have the stationary initialization, we apply the geometric-moment contraction in Theorem 1 and obtain

$$\||\beta_k - \beta^*|_s\|_q \leq \||\beta_k - \beta_k^\circ|_s\|_q + \||\beta_k^\circ - \beta^*|_s\|_q$$

$$\leq r_{\alpha,s,q}^k \||\beta_0 - \beta_0^\circ|_s\|_q + O\big(M_{s,q}\sqrt{\max\{q, s\}\alpha}\big), \tag{61}$$

which completes the proof. □

## 8.6 Functional Dependence Measure in Time Series

The functional dependence measure in time series [Wu, 2005] is a key concept in our analysis. For that we view the high-dimensional SGD iterates $\{\beta_k\}_{k\in\mathbb{N}}$ as a nonlinear autoregressive (AR) process. Recall that $\xi_k$, $k \in \mathbb{Z}$, are i.i.d. Define the shift process $\mathcal{F}_k = (\xi_k, \xi_{k-1}, \ldots)$ and its coupled version $\mathcal{F}_{k,\{l\}} = (\xi_k, \ldots, \xi_{l+1}, \xi_l', \xi_{l-1}, \ldots)$, $l \leq k$, where $\xi_l'$ is an i.i.d. copy of $\xi_l$.

The stationary sequence $\{\beta_k^\circ\}_{k\in\mathbb{Z}}$ can be represented by a functional system

$$\beta_k^\circ = h_\alpha(\xi_k, \xi_{k-1}, \ldots) = h_\alpha(\mathcal{F}_k), \quad k \geq 1, \tag{62}$$

where $h_\alpha$ is a measurable function that depends on $\alpha$ [Wiener, 1958, Wu, 2005]. Define the coupled version of $\beta_k^\circ$ by

$$\beta_{k,\{l\}}^\circ = h_\alpha(\xi_k, \ldots, \xi_{l+1}, \xi_l', \xi_{l-1}, \ldots) = h_\alpha(\mathcal{F}_{k,\{l\}}), \quad l \leq k. \tag{63}$$

The next lemma provides a bound for the functional dependence measure $\||\beta_k^\circ - \beta_{k,\{l\}}^\circ|_s\|_q$. It is later used to derive the moment bounds and the tail probability of the ASGD iterates.

**Lemma 11.** *Consider the stationary SGD sequence $\{\beta_k^\circ\}_{k\geq 1}$. Suppose that Assumptions 2 and 3 hold with some $q \geq 2$ and even integer $s \geq 2$. Then, for all $k \geq 1$ and $l \leq k$, we have*

$$\||\beta_k^\circ - \beta_{k,\{l\}}^\circ|_s\|_q^2 \leq 4\alpha^2\big(1 - 2\alpha\mu + 7(\max\{q, s\} - 1)\alpha^2 L_{s,q}^2\big)^{k-l}\Big(L_{s,q}^2\||\beta_{l-1}^\circ - \beta^*|_s\|_q^2 + M_{s,q}^2\Big).$$

*Proof of Lemma 11.* By applying Lemma 2, it follows from similar arguments as in the proof of Proposition 1 that, for each $l \leq k - 1$,

$$\||\beta_k^\circ - \beta_{k,\{l\}}^\circ|_s\|_q^2 \leq \big(1 - 2\alpha\mu + 7(\max\{q, s\} - 1)\alpha^2 L_{s,q}^2\big)^{k-l}\||\beta_l^\circ - \beta_{l,\{l\}}^\circ|_s\|_q^2. \tag{64}$$

By Assumption 3, for all $l \geq 1$,

$$\||\nabla g(\beta_{l-1}^\circ, \xi_l)|_s\|_q^2 \leq 2\||\nabla g(\beta_{l-1}^\circ, \xi_l) - \nabla g(\beta^*, \xi_l)|_s\|_q^2 + 2\||\nabla g(\beta^*, \xi_l)|_s\|_q^2$$

$$\leq 2L_{s,q}^2\||\beta_{l-1}^\circ - \beta^*|_s\|_q^2 + 2M_{s,q}^2, \tag{65}$$

which yields

$$\||\beta_l^\circ - \beta_{l,\{l\}}^\circ|_s\|_q^2 = \alpha^2\||\nabla g(\beta_{l-1}^\circ, \xi_l) - \nabla g(\beta_{l-1}^\circ, \xi_l')|_s\|_q^2$$

$$\leq \alpha^2\Big(2\||\nabla g(\beta_{l-1}^\circ, \xi_l)|_s\|_q^2 + 2\||\nabla g(\beta_{l-1}^\circ, \xi_l')|_s\|_q^2\Big)$$

$$\leq 4\alpha^2\Big(L_{s,q}^2\||\beta_{l-1}^\circ - \beta^*|_s\|_q^2 + M_{s,q}^2\Big) \tag{66}$$

Recall $\||\nabla g(\beta^*, \xi_k)|_s\|_q \leq M_{s,q}$ by Assumption 3. Therefore,

$$\||\beta_k^\circ - \beta_{k,\{l\}}^\circ|_s\|_q^2 \leq 4\alpha^2\big(1 - 2\alpha\mu + 7(\max\{q, s\} - 1)\alpha^2 L_{s,q}^2\big)^{k-l}$$

$$\cdot \Big(L_{s,q}^2\||\beta_{l-1}^\circ - \beta^*|_s\|_q^2 + M_{s,q}^2\Big). \tag{67}$$

This completes the proof. □

## 8.7 Proofs for Section 3.2

In this section, we provide the proofs for the convergence results of ASGD in Section 3.2, which can be decomposed into the proofs for Theorems 6 to 8 in Section 6.

*Proof of Theorem 7.* Recall the i.i.d. random samples $\boldsymbol{\xi}_k = (y_k, \boldsymbol{x}_k)$, the filtration $\mathcal{F}_k = (\boldsymbol{\xi}_k, \boldsymbol{\xi}_{k-1}, \ldots)$ and its coupled version $\mathcal{F}_{k,\{l\}} = (\boldsymbol{\xi}_k, \ldots, \boldsymbol{\xi}_{l+1}, \boldsymbol{\xi}'_l, \boldsymbol{\xi}_{l-1}, \ldots)$, $l \leq k$, where $\boldsymbol{\xi}'_l$ is an i.i.d. copy of $\boldsymbol{\xi}_l$. Following Wu [2005], we introduce the projection operator

$$\mathcal{P}_l[\cdot] = \mathbb{E}[\cdot \mid \mathcal{F}_l] - \mathbb{E}[\cdot \mid \mathcal{F}_{l-1}].$$

Then, we can rewrite the centered ASGD into

$$\bar{\boldsymbol{\beta}}_k^\circ - \mathbb{E}[\bar{\boldsymbol{\beta}}_k^\circ] = \frac{1}{k} \sum_{i=1}^{k} \sum_{l=0}^{i-1} \mathcal{P}_{i-l}(\boldsymbol{\beta}_i^\circ) = \frac{1}{k} \sum_{l=0}^{k-1} \sum_{i=l+1}^{k} \mathcal{P}_{i-l}(\boldsymbol{\beta}_i^\circ). \tag{68}$$

Since $\{\mathcal{P}_{i-l}(\boldsymbol{\beta}_i^\circ)\}_{i \geq l+1}$ is a sequence of martingale differences over $i$ for each $l = 0, 1, \ldots, i-1$, following Lemma D.2 in Zhang and Wu [2021] and triangle inequality, we can obtain

$$\||\bar{\boldsymbol{\beta}}_k^\circ - \mathbb{E}[\bar{\boldsymbol{\beta}}_k^\circ]|_s\|_q = \left\|\left| \frac{1}{k} \sum_{l=0}^{k-1} \sum_{i=l+1}^{k} \mathcal{P}_{i-l}(\boldsymbol{\beta}_i^\circ) \right|_s \right\|_q$$

$$\leq \frac{1}{k} \sum_{l=0}^{k-1} \left\|\left| \sum_{i=l+1}^{k} \mathcal{P}_{i-l}(\boldsymbol{\beta}_i^\circ) \right|_s \right\|_q$$

$$\leq \frac{1}{k} \sum_{l=0}^{k-1} \left( c_q \cdot s \sum_{i=l+1}^{k} \||\mathcal{P}_{i-l}(\boldsymbol{\beta}_i^\circ)|_s\|_q^2 \right)^{1/2}. \tag{69}$$

By Theorem 1 in Wu [2005], we have

$$\||\mathcal{P}_{i-l}(\boldsymbol{\beta}_i^\circ)|_s\|_q \leq \||\boldsymbol{\beta}_i^\circ - \boldsymbol{\beta}_{i,\{i-l\}}^\circ|_s\|_q. \tag{70}$$

This along with Lemma 11 and definition of $\tilde{r}_{\alpha,s,q}$ in (58) yields

$$\||\mathcal{P}_{i-l}(\boldsymbol{\beta}_i^\circ)|_s\|_q^2 \leq 4\alpha^2 \left(1 - 2\alpha\mu + 7(\max\{q,s\} - 1)\alpha^2 L_{s,q}^2\right)^l$$
$$\cdot \left( L_{s,q}^2 \||\boldsymbol{\beta}_{i-l-1}^\circ - \boldsymbol{\beta}^*|_s\|_q^2 + M_{s,q}^2 \right)$$
$$= 4\alpha^2 \tilde{r}_{\alpha,s,q}^l \left( L_{s,q}^2 \||\boldsymbol{\beta}_{i-l-1}^\circ - \boldsymbol{\beta}^*|_s\|_q^2 + M_{s,q}^2 \right). \tag{71}$$

Recall $r_{\alpha,s,q}$ in (6) and $\tilde{r}_{\alpha,s,q}$ in (58). For some constant $\omega > 0$ such that

$$\omega \leq \min\left\{ \frac{1}{\alpha}, 2\mu - 7(\max\{q,s\} - 1)\alpha L_{s,q}^2 \right\}, \tag{72}$$

we have $1 - \omega\alpha \geq 0$ and

$$r_{\alpha,s,q} \leq \tilde{r}_{\alpha,s,q} \leq 1 - \omega\alpha < 1. \tag{73}$$

Consequently, we can further bound (69) by

$$\||\bar{\boldsymbol{\beta}}_k^\circ - \mathbb{E}[\bar{\boldsymbol{\beta}}_k^\circ]|_s\|_q$$
$$\leq \frac{1}{k} \sum_{l=0}^{k-1} \left[ 4c_q s \alpha^2 (1 - \omega\alpha)^l \sum_{i=l+1}^{k} \left( L_{s,q}^2 \||\boldsymbol{\beta}_{i-l-1}^\circ - \boldsymbol{\beta}^*|_s\|_q^2 + M_{s,q}^2 \right) \right]^{1/2}$$
$$= \frac{1}{k} \sum_{l=0}^{k-1} \left[ 4c_q s \alpha^2 (1 - \omega\alpha)^l \left( L_{s,q}^2 \sum_{i=l+1}^{k} \||\boldsymbol{\beta}_{i-l-1}^\circ - \boldsymbol{\beta}^*|_s\|_q^2 + (k-l)M_{s,q}^2 \right) \right]^{1/2}$$
$$\leq \frac{1}{k} \sum_{l=0}^{k-1} \left[ 2\alpha\sqrt{c_q s}(1 - \omega\alpha)^{l/2} L_{s,q} \sqrt{\sum_{i=l+1}^{k} \||\boldsymbol{\beta}_{i-l-1}^\circ - \boldsymbol{\beta}^*|_s\|_q^2} \right]$$
$$+ \frac{1}{k} \sum_{l=0}^{k-1} \left[ 2\alpha\sqrt{c_q s}(1 - \omega\alpha)^{l/2} \sqrt{(k-l)} M_{s,q} \right] =: \mathrm{I}_1 + \mathrm{I}_2. \tag{74}$$

For the term $I_1$, it follows from Theorem 2 and expression (58) that

$$\sum_{i=l+1}^{k} \||\bar{\beta}_{i-l-1}^\circ - \beta^*|_s\|_q^2 \leq \sum_{i=l+1}^{k} \left( 6M_{s,q}^2 (\max\{q,s\} - 1)\alpha \right)$$
$$= 6\alpha(k-l)(\max\{q,s\} - 1)M_{s,q}^2. \tag{75}$$

Inserting this back into (74) gives

$$I_1 \leq \frac{2\alpha\sqrt{c_q s}L_{s,q}}{k} \sum_{l=0}^{k-1} (1-\omega\alpha)^{l/2} M_{s,q} \sqrt{6\alpha(k-l)(\max\{q,s\} - 1)}$$
$$\leq \sqrt{c_q s}L_{s,q} \cdot \frac{c_1\sqrt{\alpha}}{\sqrt{k}} M_{s,q} \sqrt{\max\{q,s\} - 1}, \tag{76}$$

for some constant $c_1 > 0$, where the last inequality is due to

$$\sum_{l=0}^{k-1} (1-\omega\alpha)^{l/2} \sqrt{k-l} \leq \sqrt{k} \sum_{l=0}^{k-1} (1-\omega\alpha)^{l/2} = O\left(\frac{\sqrt{k}}{\omega\alpha}\right). \tag{77}$$

Similarly, for some constant $c_2 > 0$,

$$I_2 \leq \frac{c_2\sqrt{c_q s}}{\sqrt{k}} M_{s,q}. \tag{78}$$

Combining the results of $I_1$ and $I_2$, we obtain the claimed inequality

$$\||\bar{\beta}_k^\circ - \mathbb{E}[\bar{\beta}_k^\circ]|_s\|_q \leq \sqrt{\frac{c_q s}{k}} \left( c_1 L_{s,q} \sqrt{\alpha} M_{s,q} \sqrt{\max\{q,s\} - 1} + c_2 M_{s,q} \right).$$

$\square$

*Proof of Theorem 6.* For the ASGD sequence $\{\bar{\beta}_k\}_{k\in\mathbb{N}}$ with arbitrarily fixed initialization $\beta_0 \in \mathbb{R}^d$ and the stationary ASGD sequence $\{\bar{\beta}_k^\circ\}_{k\in\mathbb{N}}$ with $\beta_0^\circ \sim \pi_\alpha$, we have

$$\||\bar{\beta}_k - \bar{\beta}_k^\circ|_s\|_q = \frac{1}{k} \left\| \left| \sum_{i=1}^{k} (\beta_i - \beta_i^\circ) \right|_s \right\|_q$$
$$\leq \frac{1}{k} \sum_{i=1}^{k} \||\beta_i - \beta_i^\circ|_s\|_q. \tag{79}$$

For each $1 \leq i \leq k$, it follows from the geometric-moment contraction in Theorem 1 that

$$\||\beta_i - \beta_i^\circ|_s\|_q \leq r_{\alpha,s,q}^i \||\beta_0 - \beta_0^\circ|_s\|_q. \tag{80}$$

Recall that $r_{\alpha,s,q} = 1 - 2\mu\alpha + (\max\{q,s\} - 1)L_{s,q}^2\alpha^2 < 1$ in (6). Therefore,

$$\||\bar{\beta}_k - \bar{\beta}_k^\circ|_s\|_q \leq \frac{1}{k} \cdot \frac{r_{\alpha,s,q}(1 - r_{\alpha,s,q}^k)}{1 - r_{\alpha,s,q}} \||\beta_0 - \beta_0^\circ|_s\|_q \leq \frac{1}{k} \cdot \frac{1}{1 - r_{\alpha,s,q}} \||\beta_0 - \beta_0^\circ|_s\|_q. \tag{81}$$

The desired result is achieved. $\square$

*Proof of Theorem 8.* Without loss of generality, assume $\beta^* = 0$. We use the notation (17) for the derivatives of $G$. Notice that
$$\nabla G(\beta^*) = \nabla G(0) = 0. \tag{82}$$

Consider the stationary SGD recursion

$$\beta_k^\circ = \beta_{k-1}^\circ - \alpha\nabla g(\beta_{k-1}^\circ, \xi_k), \quad k \geq 1.$$

By taking the expectation on the both sides, we obtain, for all $k \geq 1$,

$$\mathbb{E}[\nabla G(\beta_{k-1}^\circ)] = 0. \tag{83}$$

Throughout the rest of the proof, we omit the iteration index $k$ and write $\boldsymbol{\beta} = \boldsymbol{\beta}_{k-1}^{\circ}$ when no confusion is caused. For notational convenience, write $\boldsymbol{\beta} = (\beta_1, \ldots, \beta_d)^{\top}$.

A first-order Taylor expansion on $\nabla G(\boldsymbol{\beta})$ at $\boldsymbol{\beta}^* = 0$ gives

$$0 = \mathbb{E}[\nabla G(\boldsymbol{\beta})] = \nabla G(0) + \nabla^2 G(0) \mathbb{E}[\boldsymbol{\beta}] + \mathcal{R}(\boldsymbol{\beta}), \tag{84}$$

where $\nabla^2 G(0)$ is the $d \times d$ Jacobian matrix with entries defined by

$$[\nabla^2 G(0)]_{i,j} = \frac{\partial^2}{\partial \beta_i \partial \beta_j} G(\boldsymbol{\beta})\Big|_{\boldsymbol{\beta}=0}, \quad 1 \le i, j \le d, \tag{85}$$

and $\mathcal{R}(\boldsymbol{\beta})$ is the $d$-dimensional remainder defined as

$$\mathcal{R}(\boldsymbol{\beta}) = \int_0^1 \mathbb{E}\big([\nabla^2 G(t\boldsymbol{\beta}) - \nabla^2 G(0)]\boldsymbol{\beta}\big)\, dt. \tag{86}$$

The $i$-th entry of $\mathcal{R}(\boldsymbol{\beta})$ can be rewritten into

$$\mathcal{R}_i(\boldsymbol{\beta}) = \int_0^1 (1-t)\mathbb{E}\big(\boldsymbol{\beta}^{\top} \nabla^3 G_i(t\boldsymbol{\beta})\boldsymbol{\beta}\big)\, dt, \tag{87}$$

where $\nabla^3 G_i(\boldsymbol{\beta})$, $1 \le i \le d$, is a $d \times d$ matrix whose entries are

$$[\nabla^3 G_i(\boldsymbol{\beta})]_{l,r} = \frac{\partial^3}{\partial \beta_i \partial \beta_l \partial \beta_r} G(\boldsymbol{\beta}), \quad 1 \le l, r \le d. \tag{88}$$

Since $\nabla G(0) = 0$ and $\nabla^2 G(0)$ is invertible given that $\lambda_{\min}[\nabla^2 G(0)] > 0$, it follows from equation (84) that

$$\mathbb{E}[\boldsymbol{\beta}] = -[\nabla^2 G(0)]^{-1} \mathbb{E}[\mathcal{R}(\boldsymbol{\beta})]. \tag{89}$$

We only need to bound $|\mathbb{E}[\mathcal{R}(\boldsymbol{\beta})]|_s$ using Theorem 2, that is $\mathbb{E}[|\boldsymbol{\beta}_k^{\circ} - \boldsymbol{\beta}^*|_s]^2 = O(\max\{q,s\}\alpha)$ for all $k \ge 1$.

Let $\boldsymbol{v} = \boldsymbol{\beta}/|\boldsymbol{\beta}|_s$. For each $i = 1, \ldots, d$,

$$\mathbb{E}[\mathcal{R}_i(\alpha)] = \int_0^1 (1-t)\mathbb{E}[\boldsymbol{\beta}^{\top} \nabla^3 G_i(t\boldsymbol{\beta})\boldsymbol{\beta}]\, dt$$

$$= \int_0^1 (1-t)\mathbb{E}[|\boldsymbol{\beta}|_s^2 \boldsymbol{v}^{\top} \nabla^3 G_i(t\boldsymbol{\beta})\boldsymbol{v}]\, dt. \tag{90}$$

By Hölder's inequality, for $1/p + 1/q = 1$,

$$\mathbb{E}[|\boldsymbol{\beta}|_s^2 \boldsymbol{v}^{\top} \nabla^3 G_i(t\boldsymbol{\beta})\boldsymbol{v}] \le (\mathbb{E}[|\boldsymbol{\beta}|_s^{2q}])^{1/q} \cdot (\mathbb{E}(\boldsymbol{v}^{\top} \nabla^3 G_i(t\boldsymbol{\beta})\boldsymbol{v})^p)^{1/p}. \tag{91}$$

Again by Hölder's inequality,

$$\mathbb{E}[(\boldsymbol{v}^{\top} \nabla^3 G_i(t\boldsymbol{\beta})\boldsymbol{v})^p] \le d^{p(1-\frac{2}{s})} \sup_{|\boldsymbol{v}|_s=1} \mathbb{E}|\nabla^3 G_i(t\boldsymbol{\beta})\boldsymbol{v}|_s^p. \tag{92}$$

Therefore, by Theorem 2 and Lemma 10,

$$\mathbb{E}[\boldsymbol{\beta}] \lesssim M_{s,q}^2 \max\{q,s\}\alpha d^{\frac{q}{q-1}\cdot(1-\frac{2}{s})} \max_{1 \le i \le d} \|\nabla^3 G_i(\boldsymbol{\beta})\|_{\infty}, \tag{93}$$

where the matrix norm

$$\|\nabla^3 G_i(\boldsymbol{\beta})\|_{\infty} := \max_{1 \le j_1 \le d} \sum_{j_2=1}^{d} \left|\big(\nabla^3 G_i(\boldsymbol{\beta})\big)_{1 \le j_1, j_2 \le d}\right|. \tag{94}$$

Finally, given the uniform bound $\max_{1 \le i \le d} \|\nabla^3 G_i(\boldsymbol{\beta})\|_{\infty} < \infty$,

$$\mathbb{E}[\boldsymbol{\beta}] = O\Big(M_{s,q}^2 \max\{q,s\}\alpha d^{\frac{q}{q-1}\cdot(1-\frac{2}{s})}\Big), \tag{95}$$

which finishes the proof. $\qquad\square$

## 8.8 Proofs for Section 4

*Proof of Theorem 4.* By Theorem 6, we have $\||\bar{\boldsymbol{\beta}}_k - \bar{\boldsymbol{\beta}}_k^\circ|_s\|_q \lesssim 1/(k\alpha)\||\boldsymbol{\beta}_0 - \boldsymbol{\beta}_0^\circ|_s\|_q$ and consequently, it follows that

$$\mathbb{P}(|\bar{\boldsymbol{\beta}}_k - \bar{\boldsymbol{\beta}}_k^\circ|_s > z) \lesssim \frac{\||\boldsymbol{\beta}_0 - \boldsymbol{\beta}_0^\circ|_s\|_q^q}{(k\alpha z)^q}, \quad z > 0. \tag{96}$$

Then it suffices to upper bound $\mathbb{P}(|\bar{\boldsymbol{\beta}}_k^\circ - \boldsymbol{\beta}^*|_s > z)$. To this end, we first bound the dependence adjusted norm (Section 2 in Zhang and Wu [2017]) for $\{\boldsymbol{\beta}_k^\circ\}_{k\geq 1}$. By Theorem 1, elementary calculations yield

$$\||\boldsymbol{\beta}_k^\circ - \mathbb{E}[\boldsymbol{\beta}_k^\circ]|_s\|_{q,1/2-1/q} = O\left(\frac{M_{s,q}}{\alpha^{1/2-1/q}}\right).$$

Consequently, by Theorem 6.2 in Zhang and Wu [2017] and Theorem 8, we have

$$\mathbb{P}(|\bar{\boldsymbol{\beta}}_k^\circ - \boldsymbol{\beta}^*|_s > z) \lesssim \frac{(\log d)^{3q/2}(\log k)^{1+2q} M_{s,q}^q}{z^q k^{q-1}\alpha^{q/2-1}} + \exp\left(-\frac{Ckz^2\alpha^{1-2/q}}{M_{s,q}^2 \log d}\right).$$

Combining this with (96) completes the proof. □

**Theorem 9** (Theorem 3.1 in [Mies and Steland, 2023]). *Let $(\epsilon_i)_{i\in\mathbb{Z}}$ be i.i.d. random variables and $\boldsymbol{\epsilon}_k = (\epsilon_k, \epsilon_{k-1}, \ldots)$. Assume $X_k = G_k(\boldsymbol{\epsilon}_k) \in \mathbb{R}^d$ with $\mathbb{E}[X_k] = 0$ for some measurable function $G_k$. For any $k$, denote $\tilde{\boldsymbol{\epsilon}}_{k,j} = (\epsilon_k, \ldots, \epsilon_{j+1}, \tilde{\epsilon}_j, \epsilon_{j-1}, \ldots)$ with $\tilde{\epsilon}_j$ an i.i.d. copy of $\epsilon_j$. Assume there exist $\Theta > 0$ and $q > 2$, such that for all $k$,*

$$(\mathbb{E}|G_k(\boldsymbol{\epsilon}_k) - G_k(\tilde{\boldsymbol{\epsilon}}_{k,k-j})|_2^q)^{1/q} \leq \frac{\Theta}{(j\vee 1)^3}, \quad \text{for all } j \geq 0, \quad \text{and} \quad (\mathbb{E}|G_k(\boldsymbol{\epsilon}_0)|_2^q)^{1/q} \leq \Theta. \tag{97}$$

*Additionally, assume that for some $\Gamma \geq 1$,*

$$\sum_{k=2}^n (\mathbb{E}|G_k(\boldsymbol{\epsilon}_0) - G_{k-1}(\boldsymbol{\epsilon}_0)|_2^2)^{1/2} \leq \Gamma \cdot \Theta. \tag{98}$$

*If $d \leq cn$ for some $c > 0$, then on a potentially different probability space, there exist random vectors $(X_k')_{k=1}^n =^{\mathcal{D}} (X_k)_{k=1}^n$ and independent, mean zero, Gaussian random vectors*

$$Y_k^* \sim \mathcal{N}\left(0, \sum_{h=-\infty}^{\infty} \text{Cov}\left(G_k(\boldsymbol{\epsilon}_0), G_k(\boldsymbol{\epsilon}_h)\right)\right)$$

*such that*

$$\left(\mathbb{E}\max_{m\leq n}\left|\frac{1}{\sqrt{n}}\sum_{k=1}^m (X_k' - Y_k^*)\right|_2^2\right)^{1/2} \leq C\Theta\Gamma^{\frac{1}{4}}\sqrt{\log(n)}\left(\frac{d}{n}\right)^{\frac{q-2}{6q-4}},$$

*for some constant $C$ depending on $(q,c)$.*

Instead of univariate $\epsilon_i$, we apply Theorem 3.1 with vector-valued i.i.d. inputs $\boldsymbol{\xi}_i$. The theorem still applies as the proof depends only on the i.i.d. random elements and their $L^q$ bounds but not on the dimension of $\boldsymbol{\xi}_i$.

*Proof of Theorem 5.* To prove the Gaussian approximation we will apply Theorem 9 (Theorem 3.1 in Mies and Steland [2023]) with $G_k \equiv G = h_\alpha$ defined in (62) since $\boldsymbol{\beta}_k^\circ$ is stationary. We now verify the conditions (97) and (98).

Recall the functional dependence measure $\||\boldsymbol{\beta}_k^\circ - \boldsymbol{\beta}_{k,\{l\}}^\circ|_s\|_q$ introduced in Section 8.6. Throughout the proof, the $q$-th moment of the Euclidean norm is denoted by

$$\|\cdot\|_q := \||\cdot|_2\|_q.$$

Set

$$\rho_{\alpha,q}^2 := 1 - 2\alpha\mu + 7(\max\{q,2\} - 1)\alpha^2 L_{2,q}^2, \quad C_{\alpha,q} := 2\alpha\sqrt{cL_{2,q}^2 \max\{q,2\}\alpha + 1} \tag{99}$$

for some constant $c > 0$. If $c$ is chosen sufficiently large, then, by Lemma 11 and Theorem 2, for all $k \geq 1$ and $l \leq k$, we have

$$
\begin{aligned}
\|\boldsymbol{\beta}_k^\circ - \boldsymbol{\beta}_{k,\{l\}}^\circ\|_q^2 &\leq 4\alpha^2 \rho_{\alpha,q}^{2(k-l)} \left( L_{2,q}^2 \|\boldsymbol{\beta}_{l-1}^\circ - \boldsymbol{\beta}^*\|_q^2 + M_{2,q}^2 \right) \\
&\leq 4\alpha^2 \rho_{\alpha,q}^{2(k-l)} M_{2,q}^2 \left( c L_{2,q}^2 \max\{q,2\}\alpha + 1 \right) \\
&= C_{\alpha,q}^2 \rho_{\alpha,q}^{2(k-l)} M_{2,q}^2.
\end{aligned}
$$

For $\alpha \in (0, \alpha_{s_d,q})$, it follows that $\rho_{\alpha,q} < 1$. Let $l = k - j$. Then, for a sufficiently large constant $C_{\alpha,q}'$, we have

$$
\|\boldsymbol{\beta}_k^\circ - \boldsymbol{\beta}_{k,\{k-j\}}^\circ\|_q \leq C_{\alpha,q} M_{2,q} \rho_{\alpha,q}^j \leq C_{\alpha,q}' M_{2,q}(j+1)^{-3}. \tag{100}
$$

Therefore, the condition (97) holds with $\Theta = C_{\alpha,q}' M_{2,q}$. This verifies the first part of condition 97. For the second part of condition 97, by Assumption 3 and Theorem 2, for some constant $C_{\alpha,q}'' > 0$,

$$
\|h_\alpha(\boldsymbol{\xi}_0, \boldsymbol{\xi}_{-1}, \ldots)\|_q = \|\boldsymbol{\beta}_0^\circ\|_q \leq \|\boldsymbol{\beta}_0^\circ - \boldsymbol{\beta}^*\|_q + |\boldsymbol{\beta}^*|_2 \leq C_{\alpha,q}'' M_{2,q} < \infty. \tag{101}
$$

Moreover, since $\boldsymbol{\beta}_k^\circ$ is stationary, $G_k = G_{k-1} = h_\alpha$ and the left hand side of (98) is zero. Thus, condition (98) is trivially satisfied with $\Gamma = 1$.

Finally, we show that the long-run covariance matrix $\Xi = \sum_{k=-\infty}^\infty \text{Cov}(\boldsymbol{\beta}_0^\circ, \boldsymbol{\beta}_k^\circ)$ is well defined in the sense that the spectral norm $\|\Xi\|_s$ is finite. Following (63), denote

$$
\boldsymbol{\beta}_{k,\{\leq l\}}^\circ := \boldsymbol{\beta}_{k,\{\ldots,l-1,l\}}^\circ = h_\alpha(\boldsymbol{\xi}_k, \ldots, \boldsymbol{\xi}_{l+1}, \boldsymbol{\xi}_l', \boldsymbol{\xi}_{l-1}', \ldots) = h_\alpha(\mathcal{F}_{k,\{\ldots,l-1,l\}}), \quad l \leq k. \tag{102}
$$

Since $\boldsymbol{\beta}_{k,\{\leq 0\}}^\circ$ is independent of $\boldsymbol{\beta}_0^\circ$, we have

$$
\begin{aligned}
\text{Cov}(\boldsymbol{\beta}_0^\circ, \boldsymbol{\beta}_k^\circ) &= \mathbb{E}[\boldsymbol{\beta}_0^\circ \boldsymbol{\beta}_k^{\circ\top}] - \mathbb{E}[\boldsymbol{\beta}_0^\circ]\mathbb{E}[\boldsymbol{\beta}_k^{\circ\top}] \\
&= \mathbb{E}[\boldsymbol{\beta}_0^\circ \boldsymbol{\beta}_k^{\circ\top}] - \mathbb{E}[\boldsymbol{\beta}_0^\circ]\mathbb{E}[\boldsymbol{\beta}_{k,\{\leq 0\}}^{\circ\top}] \\
&= \mathbb{E}\left[\boldsymbol{\beta}_0^\circ (\boldsymbol{\beta}_k^\circ - \boldsymbol{\beta}_{k,\{\leq 0\}}^\circ)^\top\right]. \tag{103}
\end{aligned}
$$

We can rewrite the difference as a telescoping sum,

$$
\boldsymbol{\beta}_k^\circ - \boldsymbol{\beta}_{k,\{\leq 0\}}^\circ = \sum_{l=0}^\infty \left( \boldsymbol{\beta}_{k,\{\leq -l+1\}}^\circ - \boldsymbol{\beta}_{k,\{\leq -l\}}^\circ \right). \tag{104}
$$

By stationarity and (100), it follows that

$$
\left\| \boldsymbol{\beta}_{k,\{\leq -l+1\}}^\circ - \boldsymbol{\beta}_{k,\{\leq -l\}}^\circ \right\|_2 = \left\| \boldsymbol{\beta}_{k,\{-l+1\}}^\circ - \boldsymbol{\beta}_{k,\{-l\}}^\circ \right\|_2 \leq C_{\alpha,2} M_{2,2} \rho_{2,2}^{k+l+1}. \tag{105}
$$

For the spectral norm,

$$
\begin{aligned}
\left\| \text{Cov}(\boldsymbol{\beta}_0^\circ, \boldsymbol{\beta}_k^\circ) \right\|_s &= \sup_{\boldsymbol{u}, \boldsymbol{v} \in \mathbb{R}^d, |\boldsymbol{u}|_2 = |\boldsymbol{v}|_2 = 1} \mathbb{E}\boldsymbol{v}^\top \boldsymbol{\beta}_0^\circ (\boldsymbol{\beta}_k^\circ - \boldsymbol{\beta}_{k,\{\leq 0\}}^\circ)^\top \boldsymbol{u} \\
&\leq \sup_{\boldsymbol{u}, \boldsymbol{v} \in \mathbb{R}^d, |\boldsymbol{u}|_2 = |\boldsymbol{v}|_2 = 1} [\mathbb{E}(\boldsymbol{v}^\top \boldsymbol{\beta}_0^\circ)^2]^{1/2} [\mathbb{E}((\boldsymbol{\beta}_k^\circ - \boldsymbol{\beta}_{k,\{\leq 0\}}^\circ)^\top \boldsymbol{u})^2]^{1/2} \\
&\leq \|\boldsymbol{\beta}_0^\circ\|_2 \|\boldsymbol{\beta}_k^\circ - \boldsymbol{\beta}_{k,\{\leq 0\}}^\circ\|_2, \tag{106}
\end{aligned}
$$

where the first inequality is by Cauchy-Schwarz and the last inequality uses $(\boldsymbol{u}^\top \boldsymbol{\beta}_0^\circ)^2 \leq |\boldsymbol{\beta}_0^\circ|^2$ with $|\boldsymbol{u}|_2 = 1$. This, along with $M_{2,2} < \infty$ (Assumption 3) yields

$$
\left\| \text{Cov}(\boldsymbol{\beta}_0^\circ, \boldsymbol{\beta}_k^\circ) \right\|_s \leq \|\boldsymbol{\beta}_0^\circ\|_2 \|\boldsymbol{\beta}_k^\circ - \boldsymbol{\beta}_{k,\{\leq 0\}}^\circ\|_2 \leq C_\alpha' \rho_{\alpha,2}^k, \tag{107}
$$

for some constant $C_\alpha' > 0$. As a direct consequence,

$$
\|\Xi\|_s \leq \left\| \mathbb{E}[\boldsymbol{\beta}_0^\circ \boldsymbol{\beta}_0^{\circ\top}] \right\|_s + 2 \sum_{k=1}^\infty C_\alpha' \rho_{\alpha,2}^k < \infty. \tag{108}
$$

This completes the proof.

$\square$

