# OpenReview forum: "Statistical Guarantees for High-Dimensional Stochastic Gradient Descent"
_NeurIPS.cc/2025/Conference — NeurIPS 2025 poster_

### Official Review · Reviewer_WFcH · 2025-06-11

**Clarity:** 4
**Significance:** 4
**Originality:** 4
**Rating:** 5
**Confidence:** 4

**Summary:**

The paper derives new convergence results for SGD and ASGD in high-dimensional settings with constant learning rate.

The key theory of the paper is to derive generalized moment convergence of (A)SGD and apply the coupling technique in nonlinear time series analysis to show the high probability concentration bound of ASGD.

**Questions:**

The only question I have is whether the convergence of ASGD in Theorem 3 and the high-probability concentration bound in Theorem 4 can be extended to $\ell^s$ norms. It would be helpful if the authors would remark on the technical reasons why this analysis is not provided.

**Ethical Concerns:**

["NO or VERY MINOR ethics concerns only"]

**Final Justification:**

After reading the author’s response, I find these results reasonable, and I will keep my score unchanged.

**Limitations:**

Again, I generally like the presentation of this work and I am not aware of any limitations.

**Quality:**

4

**Strengths And Weaknesses:**

**Strength:**

The paper is well written and with clear technical contributions in understanding the convergence of SGD and ASGD in high dimensional settings.

Theorem 1 shows the convergence of SGD to a stationary distribution. In particular, the analysis is based on proving that for sufficiently small constant learning rates, the initial distance between two SGD sequences decreases exponentially fast. This is, to my knowledge, a new technique, with which the authors extend the results in previous works (e.g. [Li et al. 2024b]) to high-dimensional settings and general norms.

Theorem 2 and 3 show the convergence of (A)SGD in general moments. Specifically, Theorem 2 shows the last-iterate convergence of SGD in $\ell^s$ norm; Theorem 3 shows the convergence of ASGD in $\ell^\infty$ norm. This recovers some best known result (e.g., [Wainwright 2019]) as a special case.

Theorem 4 derives a Fuk-Nagaev-type inequality of ASGD for high-probability concentration bound in the infinity norm. This leads to a sharp high-probability bound on the infinity norm in certain regimes of the gradient noise.

The results are presented in a clean and self-contained way. The proof sketch in section 6 highlights the technical contributions and is very helpful in understanding the techniques.

I have also checked the analysis and found the proofs to be correct.

**Weakness:**

I appreciate the theoretical results, presentation, and proof techniques in this paper. I did not spot any particular weakness.


**Updated review:**
Thank for the author’s response! I find these results reasonable, and I will keep my score unchanged.

---

> ### Author Rebuttal · Authors · 2025-07-30
>
> We are extremely grateful for your strong endorsement and careful review of our proofs. Your support has been invaluable in framing the impact and future directions of our analysis. Please see below our response to your question on the $\ell^s$-norms.
>
> **The only question I have is whether the convergence of ASGD in Theorem 3 and the high-probability concentration bound in Theorem 4 can be extended to $\ell^s$ norms. It would be helpful if the authors would remark on the technical reasons why this analysis is not provided.**
>
> **Answer:** Thank you so much again for this great question! Indeed, all of our derived theory, including the convergence of ASGD in Theorem 3 and the high-probability concentration bound in Theorem 4, work for general $\ell^s$-norms with $s\ge2$. In fact, we did not provide them in the current version due to the concern of space. According to your suggestion, we will add these results to the Appendix in the revision for the completeness of our work. We would like to summarize them over here.
>
> **Theorem 3 (Convergence of ASGD in $\ell^s$-norm).** *Consider the ASGD sequence $\\{\bar{\boldsymbol{\beta}}\_k\\}\_{k \geq 1}$. Suppose that Assumptions 1 and 2 hold with some $q \geq 2$ and even integer $s \geq 2$, the conditions in Theorem 7 hold and the learning rate satisfies $\alpha \in (0, \alpha\_{s\_d, q})$ with $\alpha\_{s\_d, q}$ defined in (5). For any $k \geq 1$ and some universal constants $C\_1,C\_2,C\_3 > 0$,*
>
> \begin{equation}
>     \\||\bar{\boldsymbol{\beta}}\_k - \boldsymbol{\beta}^*|\_s\\|\_q  \leq C\_1 \{\underbrace{\sqrt{\frac{c\_qs}{k}}M\_{s,q}(L\_{s,q}\sqrt{\alpha\max\\{q,s\\}} + 1)}\_{\text{\small{stochastic variance}}}\}  +  C\_2\{\underbrace{\frac{1}{k(1-r\_{\alpha,s,q})}\\||\boldsymbol{\beta}\_0-\boldsymbol{\beta}\_0^{\circ}|\_{\infty}\\|\_q}\_{\text{\small{initialization bias}}}\} + C\_3\{\underbrace{M\_{s,q}^2\max\\{q,s\\}\alpha d^{\frac{q}{q-1}\cdot(1-\frac{2}{s})}}\_{\text{\small{bias of constant learning rate}}}\}.
> \end{equation}
>
> **Theorem 4 (Fuk-Nagaev inequality in $\ell^s$-norm).** *Under the conditions of Theorem 3*, for any $z > 0$, we have*
>
> \begin{equation}
> \mathbb{P} (|\bar{\boldsymbol{\beta}}\_{k} - \boldsymbol{\beta}^{*}|\_s > z) \lesssim \frac{\\||\boldsymbol{\beta}\_{0} - \boldsymbol{\beta}\_{0}^{\circ}|\_s\\|\_q^{q}}{(k\alpha z)^{q}} + \frac{(\log d)^{\frac{3q}{2}} (\log k)^{1 + 2 q} M\_{s, q}^{q}}{z^{q} k^{q - 1} \alpha^{q/2 - 1}} + \exp\left(- \frac{C k z^{2} \alpha^{1 - 2/q}}{M\_{s, q}^{2} \log d}\right),
> \end{equation}
>
> *where the constants in $\lesssim$ are independent of $k,d,s$ and $\alpha$.*
>
>
> Notably, since the geometric-moment contraction (GMC) (Theorem 1) is developed for general $\ell^s$-norms, all the subsequent analysis based on the GMC property is also derived for general $s$. Considering our primary interest is to provide convergence in $\ell^{\infty}$-norm by choosing $s=s_d\approx\log(d)$ such that $\ell^{s\_d}$-norm approximate $\ell^{\infty}$-norm, we will keep the $\ell^{\infty}$-type results in the main text and add Theorems 3* and 4* to the Appendix according to your suggestion.

---

> > ### Comment · Reviewer_WFcH · 2025-08-05
> >
> > We thank the reviewer for the response and for sharing the new results. They look reasonable to me.
> >
> > Also, as per the discussion above with Reviewer BCC1, an additional assumption of coercivity or bounded sub-level set will be sufficient to fix the proof of Lemma 4. This also looks fine to me.
> >
> > Therefore, I keep my score unchanged.

---

### Official Review · Reviewer_BCC1 · 2025-06-26

**Clarity:** 2
**Significance:** 2
**Originality:** 2
**Rating:** 4
**Confidence:** 2

**Summary:**

This paper presents statistical guarantees for SGD in smooth and strongly convex setting for SGD and averaged SGD (ASGD). More precisely, the authors are interested in proving moment convergence bounds and assume strong convexity and expected smoothness in $\ell^s$ and $L^q$ norms, in a sense that is made precise in the paper. Under these assumptions, the authors prove results on the convergnence of SGD to its stationary distribution as well as moment convergence of SGD, in both cases under additional restrictions on the learning rate. Moment convergence bounds are also obtained for ASGD. In the case of ASGD, high probability convergence bounds are also obtained, through the derivation of a so-called Fuk-Nagaev type inequality. The authors also analyze the dependence of the learning rate constraints on the parameters of the problem (in particular the dimension) and provide sketches of proofs of their main results. The proofs rely in particular on the adaptation of a coupling technique to online SGD.

**Questions:**

- Your theoretical results rely on the assumption that $s$ is even. It is clear from the proof that this assumption simplifies some computations, however, it does not feel very intuitive. Could you add a discussion about potentially removing this condition and how critical it is?
- Line 123, you mention that using constant learning rates prevail in practical scenarios, but a large number of works seem to use learning rate schedules to train modern models. Can you be more precise about when constant learning rates are used? Moreover, how essential is the constant learning rate assumption in your work? Could all your results be easily generalized the non-constant learning rates?
- Could you provide more justification as to why it is essential to bound the quantities $\Vert |\beta_k - \beta^\star|_s \Vert q$? Why is it necessary to have such bounds for all $s$ and $q$?
- As I understand, Lemma is not used in the proofs, is it useful to present it in the main text? Moving it to the appendix could allow for more space for qualitative discussions of the results.

**Ethical Concerns:**

["NO or VERY MINOR ethics concerns only"]

**Final Justification:**

The paper is well-written and present new results regarding statistical guarantees for SGD. The main analysis is based on a $s$-convexity assumptions.

During the review process, a mistake was found in the proof of lemma 4, showing that the $s$-convexity alone does not ensure the existence of a global minimum.
After several (and very positive) discussions with the authors and other reviewers, this problem was fixed.

I am now confident that this paper is above the bar of acceptance, after fixing these issues.

For this reason, I have raised my initial score.

**Limitations:**

Some limitations are discussed by the authors, in particular the fact that their analysis apply only in smooth and strongly-convex settings.

**Quality:**

3

**Strengths And Weaknesses:**

**Strengths:**

- The sketches of proofs provide a good idea of how the proof work without harming the readability of the paper.
- Some of the bounds extend known results in the mean square error case.
- The introduction of coupling technique is a powerful research direction, it is possible that some elements introduced in the paper can be applied in other settings.
- New high-probability bounds (Nagaev type inequality) are derived for ASGD

**Weaknesses:**

Here are some potential weaknesses, please correct me if I am wrong.

- About assumption 1. The notation $\beta^s$ is not clear when $\beta \in R ^d$. We can see from the proof that the product is meant element-wise but it should be mentioned. Because of this element-wise product, Assumption 1 should be discussed a bit more, especially whether it has been used in other works. Additionally, the proof of Lemma 4 looks a bit suspicious. Indeed, the first integral features $\int \langle \beta, \dots$, but a few lines later it is transformed into $\int \langle \beta^{s-1} ,\dots$ without justification.
- About Assumption 2. It is mention that the constants $L_{s,q}$ and $M_{s,q}$ may depend on $d$ through their dependence on $s$. However, it seems to me that they also have a intrinsic dependence on $d$ that should be discussed. For instance, is there a link between $L_{s,q}$ and $L_{2,q}$ in the worst case?
- In the introduction, the paper is partially justified by the fact that SGD iterates are not stationary when the learning rate is large, but we see in Section 5 that the assumption impose at least $\alpha$ to be of order $1 / (d^2 \log d)$, which is small in high dimensional settings. In that regard, can we say that the paper apply to learge learning rates.
- The results only rely on smoothness and strong convexity assumptions. Moreover, these two assumptions are formulated for $\ell^s$ norms, so they are different from the standard ones.

I am not familiar enough with this literature to fully assess the technical novelty of the results.

**Other remarks**

- In the introduction, the assumption under which the results hold should be mentioned alongside the informal presentation of the result, to help the readability.
- In some sections (eg, related works), the reference formatting and the high number of citations make the paragraphs hard to read.
- The acronym GMC is used before being defined.
- Line 264, the proof is refered to Section 2 in the appendix but it should be Section 8.4.

---

> ### Author Rebuttal · Authors · 2025-07-30
>
> Thank you so much for your detailed critique and for highlighting opportunities to improve our presentation. Your insights have made the manuscript clearer and more accessible. Please see below the detailed responses to your concerns.
>
> **1. About assumption 1. The notation $\beta^s$ is not clear when $\beta\in\mathbb{R}^d$. We can see from the proof that the product is meant element-wise but it should be mentioned. Because of this element-wise product, Assumption 1 should be discussed a bit more, especially whether it has been used in other works. Additionally, the proof of Lemma 4 looks a bit suspicious. Indeed, the first integral features $\int\langle\beta,\ldots$, but a few lines later it is transformed into $\int\langle\beta^{s-1},\ldots$ without justification.**
>
> We sincerely apologize for the confusion. We have changed the notation $\boldsymbol{\beta}^s$ to $\boldsymbol{\beta}^{\odot s}$ to denote element-wise product. The confusion in Lemma 4 proof is also fixed. We indeed need to additionally assume Assumption 1 to hold for $s=2$ to ensure the existence of global optimum. In the revision, we will fix this. In fact, there is no apparent relationship between the $\ell^s$ and $\ell^2$-type strong convexity assumptions in general cases, since different norms are involved. We therefore study the linear regression as an example in Section 8.2 to compare these two assumptions.
>
> **2. About Assumption 2. It is mention that the constants $L\_{s,q}$ and $M\_{s,q}$ may depend on $d$ through their dependence on $s$. However, it seems to me that they also have an intrinsic dependence on $d$ that should be discussed. For instance, is there a link between $L\_{s,q}$ and $L\_{2,q}$ in the worst case?**
>
> This is indeed a very good observation. We strongly agree that both $L\_{s,q}$ and $M\_{s,q}$ have an intrinsic dependence on $d$, which we have carefully discussed in Lemmas 6 and 7 in the Appendix, respectively. We apologize for not explicitly pointing to these two lemmas below Assumption 2, which shall be added in the revision. As we can see from the results of Lemma 6, $L\_{s,q}$ differs from $L\_{2,q}$ up to logarithmic factors $\sqrt{\log(d)}$ and $\log(d)$ when the noises are sub-Gaussian and sub-exponential, respectively. When the noise only has finite $q$-th moment, then $L\_{s,q}$ differs from $L\_{2,q}$ by a factor $d^{1/(2q)}$. We would like to emphasize that all the theory in this work includes $s=2$ as a special case. We have included a detailed discussion regarding the relationship between $\ell^s$ and $\ell^2$ norms in our reply to your Comment 4.
>
> **3. In the introduction, the paper is partially justified by the fact that SGD iterates are not stationary when the learning rate is large, but we see in Section 5 that the assumption impose at least $\alpha$ to be of order $1/(d^2\log d)$, which is small in high dimensional settings. In that regard, can we say that the paper apply to large learning rates.**
>
> This is in fact a typo. In the introduction, we meant a constant learning rate that does not decay in the number of iterations $k$. Indeed, convergence requires $\alpha$ to decay with the dimension of the problem. We do agree that in a high-dimensional setting one can call the learning rate small.
>
> **4. The results only rely on smoothness and strong convexity assumptions. Moreover, these two assumptions are formulated for $\ell^s$ norms, so they are different from the standard ones.**
>
> We greatly appreciate your comment on our assumptions. Indeed, our results only rely on smoothness and strong convexity assumptions. This is actually why our proposed geometric-moment contraction (GMC) is very useful in SGD algorithms, which provides convergence to the stationary distribution without any restriction on the initialization, or any limitations on the data distributions (e.g., we do not require sub-Gaussian gradients that is frequently assumed in literature), except a mild condition on the finite $q$-th moment of the stochastic gradient at the global optimum for some $q \geq 2$. This GMC property can lead to fruitful (non)-asymptotic convergence behaviors of SGD iterates, including the in-expectation and in-probability convergence.
>
> Regarding the two assumptions in $\ell^s$-norm, we would like to clarify that although we suggest large $s$ with $s\approx\log(d)$ to make $\ell^s$-norm approximate $\ell^{\infty}$-norm, all of our derived theory also works for small $s$, including $s=2$ as a special case. For $s=2$, Assumptions 1 and 2 reduce to the standard strong convexity and stochastic Lipschitz continuity for $\ell^2$-norms, and the moment convergence in Theorem 2 reduces to $(\mathbb{E}|\boldsymbol{\beta}\_k-\boldsymbol{\beta}^*|\_2^2)^{1/2}=O(M\_{2,q}\sqrt{\max\\{q,2\\}\alpha}+r\_{\alpha,2,q}^k|\boldsymbol{\beta}\_0-\boldsymbol{\beta}\_0^{\circ}|\_2)$, where the dependency on the dimension $d$ will be reflected by $M\_{2,q}$, $\alpha$ and $L\_{2,q}$ included in the contraction speed $r\_{\alpha,2,q}$. Moment convergence for ASGD in Theorem 3 and tail probability inequality in Theorem 4 will follow similarly. When $s>2$, indeed our assumptions are different from the standard ones. Such condition is required for a general $s$ since our primary goal is to derive high-dimensional convergence in $\ell^{\infty}$-norm, which can be approximated by $\ell^s$-norm with $s\approx\log(d)$.
>
> **5. Your theoretical results rely on the assumption that $s$ is even. It is clear from the proof that this assumption simplifies some computations, however, it does not feel very intuitive. Could you add a discussion about potentially removing this condition and how critical it is?**
>
> We sincerely appreciate your feedback. Indeed, assuming $s$ is an even integer is just a convenient way to define the map
>
> \begin{equation}
> \mathbf{v}\mapsto\mathbf{v}^{s-1}=(|v_1|^{s-2}v_1,\ldots,|v_d|^{s-2}v_d),
> \end{equation}
>
> without worrying about fractional powers of negative numbers. Without this assumption, our result will be unreadable and the conditions will be messy. In fact, the most frequently adopted $s$ is 2 and $s\_d$ in Eq. (7) to measure the convergence of algorithms, that is, $\ell^2$ and $\ell^{\infty}$ norms. As $s\rightarrow\infty$ to approximate $\ell^{\infty}$ norms, $s$ and $s+1$ will not differ too much. We will add this discussion in the revision.
>
> **6. Line 123, you mention that using constant learning rates prevail in practical scenarios, but a large number of works seem to use learning rate schedules to train modern models. Can you be more precise about when constant learning rates are used? Moreover, how essential is the constant learning rate assumption in your work? Could all your results be easily generalized the non-constant learning rates?**
>
> Thanks so much for your questions on the learning rates. We do agree that learning rate schedules are widely used. Constant learning rates are among the most effective schedules; see for example, Krizhevsky et al. (2012), Iiduka (2022) and the tutorial of PyTorch SGD "torch.optim". More precisely, when one needs to fix the initialization without any prior knowledge (e.g., the initial point could be very far from the optimum), a constant learning rate can speed up the convergence of the algorithm to a stationary stage. See for example, the very recent work by Defazio et al. (2024) proposed optimal learning schedules with learning rate warm-up and rapid learning rate annealing near the end of training. A constant learning rate can be used in the warm-up stage, where our proposed theory applies. We will cite these references in the revision. Indeed, our theory can be generalized to non-constant cases such as linear decaying learning rates, which however would require heavy machinery, and we will consider it as a potential future direction.
>
> **7. Could you provide more justification as to why it is essential to bound the quantities $\\||\boldsymbol{\beta}\_k-\boldsymbol{\beta}^*|_s\\|\_q$? Why is it necessary to have such bounds for all $s$ and $q$?**
>
> We are truly sorry for this confusion. In fact, we only need to bound the quantity $\\||\boldsymbol{\beta}\_k-\boldsymbol{\beta}^*|\_s\\|\_q$ for some $s$ and $q$ that are of interest. We allow for a large range of possible $s, q$ as indicated in our theorems. We have revised the statement in the revision to avoid confusion.
>
> **8. As I understand, Lemma is not used in the proofs, is it useful to present it in the main text? Moving it to the appendix could allow for more space for qualitative discussions of the results.**
>
> Thank you very much for your suggestion. We assume you refer to Lemma 3. Indeed, Lemma 3 is a consequence of the moment inequality in Lemma 2 and is of independent interest. We will move it to the Appendix in the revised version.

---

> > ### Comment · Reviewer_BCC1 · 2025-08-01
> > **Thank you for your answer**
> >
> > Thank you very much for you rebuttal and clarifying some concerns.
> >
> > In particular, thank you for the correcting the proof of Lemma 4. As I understand it, you need to assume Assumption 1 with $s=2$, therefore is it correct that it reduces to the usual strong convexity assumption and, hence, to a known result?
> >
> > Moreover, it raises another question: in some of your other results, Assumption 1 is made for an arbitrary s, however, the existence of a global minimizer seems to be implicitly assumed (in Theorems 2 and 3 for instance). How is the existence of the global minimizer ensured in these cases? Do you need further assumptions for a global minimizer to exist in these cases?

---

> > > ### Author Response · Authors · 2025-08-01
> > > **Thank you for your follow-up question!**
> > >
> > > Thanks so much for your comment and the great follow-up question!
> > >
> > > Indeed, we need to assume Assumption 1 with $s=2$ throughout our paper, which reduces to the usual strong convexity (the known result that guarantees the existence of a global minimizer). To clarify, all the results in the paper needs Assumption 1 to hold with $s=2$. With this assumption alone, all the results that are shown in the current version in terms of $\ell^s$ norm will hold with $s=2$, such as the moment convergence of SGD (Theorem 2) and ASGD (Theorem 3). If in addition, Assumption 1 holds for an arbitrary $s$, then the results such as Theorems 2 and 3 will additionally hold for this certain $s$ as well.
> > >
> > > We sincerely apologize again for the confusion! In the revision, we will explicitly state that throughout the paper, we assume Assumption 1 holds with $s=2$.

---

> > > > ### Comment · Reviewer_BCC1 · 2025-08-03
> > > > **Thank you for your answer**
> > > >
> > > > Thank you for clarifying this point and adding the strong convexity assumption in all the results.
> > > >
> > > > As I understand it, the results require two strong convexity assumptions to hold (s=2 and another value of s, like $2 \log(d)$). While this makes the assumptions stronger, I believe that the authors will accordingly clarify it in the paper.
> > > >
> > > > How restrictive do you think the assumptions become with the added strong convexity?
> > > > It might also be important to discuss examples of functions satisfying both assumptions.
> > > >
> > > > I will consider changing my score after discussion with the other reviewers.

---

> > > > > ### Comment · Reviewer_WFcH · 2025-08-03
> > > > >
> > > > > Thanks for your discussions!
> > > > >
> > > > > I am a bit confused why the strong convexity is required here. For fixing the Lemma 4, I personally prefer to directly assume a bounded sub-level set, which is also a quite common assumption and is much weaker than strong convexity.
> > > > >
> > > > > Additionally, I completely agree with Reviewer BCC1 that adding some simple analytical examples helps to show how the assumptions and the results of the paper are essentially different and generalized from previous work.
> > > > >
> > > > > -Reviewer WFcH

---

> > > > > ### Author Response · Authors · 2025-08-04
> > > > > **Reply to Reviewers BCC1 and WFcH on strong convexity assumption(s)**
> > > > >
> > > > > Thank you both so much for your comments! Indeed as suggested by Reviewer WFcH, to ensure the existence of the global optimum, the essential assumption we need is the coercivity condition
> > > > > \begin{equation*}
> > > > >     \lim\_{|\boldsymbol{\beta}|\_s\rightarrow\infty}G(\boldsymbol{\beta}) = \infty,
> > > > > \end{equation*}
> > > > > which directly implies the bounded sub-level set. We strongly agree that this condition is much weaker than the additional $\ell^2$ strong convexity and is more frequently adopted in the literature. We would like to fix our Lemma 4 by posting this coercivity condition instead.
> > > > >
> > > > > We also sincerely appreciate your suggestions for us to provide examples that satisfy our $\ell^s$ strong convexity assumption. Specifically, in Lemma 5 in the Appendix, we quantify how restrictive the additional $\ell^s$ strong convexity with $s>2$ is compared to the standard strong convexity ($s=2$), and we would like to add a more detailed discussion in the revision. In fact, there is no apparent relationship between these two strong convexity conditions in general cases since different norms are involved. We therefore study the linear regression as an example in Lemma 5. For the linear regression setting in Section 8.2, we have defined $\lambda\_{\min}^{(s)}=\inf\_{\|\mathbf{v}\|\_s=1}\langle\mathbf{v}^{s-1},\Sigma\mathbf{v}\rangle$. When $s=2$, $\lambda\_{\min}^{(2)}=\lambda\_{\min}(\Sigma)\ge \lambda\_{\min}^{(s)} \ge\min\_i(\Sigma\_{ii}-\sum\_{j\neq i}|\Sigma\_{ij}|)$. The rightmost "Gershgorin gap" is a universal lower bound for every $s$. The lower bound is non-trivial if $\Sigma$ is sufficiently diagonally dominant. This $\lambda\_{\min}^{(s)}$ can also be connected with the smallest eigenvalue of tensors introduced in Lim (2006) (CAMSAP '05) with a long version on arXiv:math/0607648 [math.SP]. We will add a sentence below Assumption 1 to point readers to this new discussion in the Appendix.

---

> > > > > > ### Comment · Reviewer_BCC1 · 2025-08-04
> > > > > > **Thank you**
> > > > > >
> > > > > > Thank you for your answer.
> > > > > >
> > > > > > I agree with the remarks above that adding a coercivity or bounded sub-level set is a better and easy way to ensure the existence of the local minimum.
> > > > > > In that case, the paper should make it clear that the $s$-convexity assumption alone does not imply the existence of a global minimum when $s>2$.
> > > > > >
> > > > > > Can it be an issue in some parts of your theory if the global minimiser is not unique? Or do you need to make stronger assumptions so that it is the case?
> > > > > >
> > > > > > Finally, thank you for adding a discussion of the examples and how $s$-convexity compares to strong convexity.

---

> > > > > > > ### Author Response · Authors · 2025-08-04
> > > > > > > **Thanks a lot for your comment!**
> > > > > > >
> > > > > > > Thanks a lot for your comment!
> > > > > > >
> > > > > > > We completely agree with your suggestion. In the revision, we will make it clear that the $s$-strong-convexity assumption alone does not imply the existence of a global minimum when $s>2$. We will post the coercivity assumption instead to ensure the existence.
> > > > > > >
> > > > > > > Regarding the uniqueness of the global minimum, in fact the $s$-strong-convexity alone (current Assumption 1) is enough to ensure this. There is no need to add any additional assumption. Please kindly refer to Lines 559-562 in our Appendix for the proof of Lemma 4 for details. Therefore, all our current theory hold accordingly.

---

> > > > > > > > ### Comment · Reviewer_BCC1 · 2025-08-05
> > > > > > > > **Thank you**
> > > > > > > >
> > > > > > > > Thank you for your answer.
> > > > > > > >
> > > > > > > > I am now satisfied with the way Lemma 4 is fixed (and indeed uniqueness is ensured). Thank you for taking the time to answer my concerns on this, I am confident that the paper will be accordingly changed.
> > > > > > > >
> > > > > > > > For these reasons, I will raise my score.
> > > > > > > >
> > > > > > > > Good luck with your paper.

---

### Official Review · Reviewer_sNN4 · 2025-06-27

**Clarity:** 3
**Significance:** 3
**Originality:** 3
**Rating:** 5
**Confidence:** 4

**Summary:**

This work provides statistical averaged and last-iterate guarantees for high dimensional stochastic gradient descent. The authors demonstrate a generalized moment contraction, leading to last-iterate convergence to a stationary distribution around the true parameters, and $q$-th moment convergence of the iterates. Of independent interest is a novel sharp Fuk-Nagaev inequality for bounding the distance in $L_{\infty}$ norm between the estimate and true model parameters.

**Questions:**

Overall, the paper is well written and gives a series of guarantees to an important area at the intersection of high-dimensional statistics.
I have a few questions concerning the asymptotic results:
* Is there a fundamental difficulty derived from the high-dimensional regime that renders demonstrating strong asymptotic consistency of the parameters difficult?
* Is it possible to show a central limit theorem for the iterates of the stochastic approximation process?
* How strong is the $\ell_s$ strong convexity assumption in relationship to the $\ell_2$ strong convexity?

**Ethical Concerns:**

["NO or VERY MINOR ethics concerns only"]

**Final Justification:**

The authors provide a well-written and thorough treatment of the statistical properties of stochastic gradient descent in the high-dimensional regime.

My main concerns were about the strength of the geometric moment contraction result and its connection to asymptotic behavior of the iterates. In their rebuttal, the authors gave a detailed explanation showing that this condition ensures both convergence to a *stationary distribution* and *Gaussian approximation* for the iterates. They also clarified the novelty of extending these results to the high-dimensional setting.

For these reasons, I maintain my positive evaluation (5 – Accept).

**Limitations:**

Yes.

**Quality:**

3

**Strengths And Weaknesses:**

Strengths:
* The paper is clearly written and technically sound, covering the settings of both last-iterate and averaged-iterate behavior with an extensive section in the appendix for the application of these methods to the important application of high-dimensional regression.
* The techniques for demonstrating Geometric-Moment contraction are new, and more robust when compared to previous results in the literature, providing a bridge to $L_{\infty}$ guarantees.
* The Fuk-Nagaev concentration inequality is novel and has uses in many stochastic approximation settings.

Weaknesses:
* The asymptotic results largely focus on the convergence in moments rather than the stronger notions of convergence almost surely or in distribution.*
* The connection between geometric moment contraction and convergence to the stationary distribution is not entirely clear.

---

> ### Author Rebuttal · Authors · 2025-07-30
>
> We greatly appreciate your positive assessment of our technical innovations. Your constructive feedback has greatly contributed to enhancing the quality of our work. Please see our responses to your concerns and questions below.
>
> **1. The asymptotic results largely focus on the convergence in moments rather than the stronger notions of convergence almost surely or in distribution.**
>
> We greatly appreciate your comment. Indeed, our first main theorem proves the geometric-moment contraction (GMC) of the SGD iterates (Theorem 1). However, we would like to emphasize that this GMC property further implies the convergence of the SGD iterates to a unique stationary distribution $\pi_{\alpha}$ as the iteration number $k$ grows (see below Eq. (6)) that is independent of the initialization. Related to your next comment, the key connection between the GMC and the convergence to the stationary distribution is based on the construction of "backward iteration" which was first introduced by Propp and Wilson (1996) and later extended to the iterated random function by Wu and Shao (2004). Please kindly refer to our reply to your next comment for details.
>
> Actually, based on this GMC property, we can derive very fruitful (non)-asymptotic results beyond the moment convergence of the estimation error in Theorems 2 and 3. In Theorem 4, we provide a tail probability inequality, which facilitates the probabilistic guarantees of the high-dimensional SGD algorithms. Moreover, related to your Comment 4, the GMC property does imply a central limit theorem for the SGD iterates. Please see our reply to your Comment 4 for details.
>
> Note that convergence in probability implies that there exists a subsequence that converges almost surely. However, we admit that deriving almost sure convergence of stochastic approximation in the high-dimensional settings is difficult. It is also rarely investigated in the literature.
>
> **2. The connection between geometric moment contraction and convergence to the stationary distribution is not entirely clear.**
>
> We sincerely apologize for this confusion. The main idea is to view the SGD recursion as an iterated random function and then use the idea "backward iteration" introduced by Propp and Wilson (1996) to find the limiting stationary distribution.
>
> Specifically, we rewrite the SGD recursion into an iterated random function as follows
> \begin{equation*}
> \boldsymbol{\beta}\_k=\boldsymbol{\beta}\_{k-1} - \alpha \nabla g(\boldsymbol{\beta}\_{k-1},\boldsymbol{\xi}\_k)=:f\_{\boldsymbol{\xi}\_k}(\boldsymbol{\beta}\_{k-1}) = f\_{\boldsymbol{\xi}\_k}\circ \cdots\circ f\_{\boldsymbol{\xi}\_1}(\boldsymbol{\beta}\_0),
> \end{equation*}
> for $k=1,2,\ldots$. One cannot directly find a limit for this forward iteration because by the Markov property, given the present position of the chain, the conditional distribution of the future does not depend on the past (see Diaconis and Freedman (1999) for the ergodicity of this forward iteration). As such, we define a backward iteration
> \begin{equation*}
> \mathbf{z}\_k(\boldsymbol{\beta}\_0) = \mathbf{z}\_{k-1}(f\_{\boldsymbol{\xi}\_k}(\boldsymbol{\beta})) = f\_{\boldsymbol{\xi}\_1}\circ \cdots\circ f\_{\boldsymbol{\xi}\_k}(\boldsymbol{\beta}\_0),
> \end{equation*}
> for $k=1,2,\ldots$. By Theorem 2 in Wu and Shao (2004), we can show that the distribution of $\mathbf{z}\_k$ is the same as the one of $\boldsymbol{\beta}\_k$, and there exists a unique random vector $\mathbf{z}\_{\infty}\in\mathbb{R}^d$ that is independent of $\boldsymbol{\beta}\_0$, such that for any $\boldsymbol{\beta}\_0\in\mathbb{R}^d$, $\mathbf{z}\_k\rightarrow\mathbf{z}\_{\infty}$ almost surely as $k\rightarrow\infty$. Hence, one can show that $\boldsymbol{\beta}\_k$ also converges to $\mathbf{z}\_{\infty}$ in distribution. We will add this detailed explanation to our revision.
>
> **3. Is there a fundamental difficulty derived from the high-dimensional regime that renders demonstrating strong asymptotic consistency of the parameters difficult?**
>
> We appreciate this insightful question! Indeed, a major challenge in establishing strong (non)-asymptotic consistency (e.g., in moment or in distribution) of SGD iterates in the high-dimensional regime lies in modeling both non-stationarity and dependencies across high-dimensional components. Traditional methods for constant learning-rate SGD typically establish convergence only in a probability measure (e.g., Wasserstein-2 distance in Dieleveut et al. (2020); Merad and Gaïffas (2023); Huo et al. (2023)), which cannot deliver refined non‑asymptotic control over high‑order moments or tail behavior, especially once $d$ grows large. This is because the constants in their theory depend on $d$ in an extremely complicated manner that are quite uneasy to work out.
>
> In our work, we propose GMC with a clean form by adapting coupling techniques from high‑dimensional nonlinear time series to show that any two SGD paths contract in $\ell^s$-norm, thereby enabling precise control of the $q$-th moment in general $\ell^s$ ($\ell^{\infty}$) norms. As a key technique, we derive sharp high‑dimensional moment inequalities that carefully track how the dimension $d$ affects the range of learning rate $\alpha$ and how it enters into the contraction speed. Moreover, our probabilistic guarantees in high dimensions are even more challenging, since we do not require light-tail assumptions (e.g. sub-Gaussian) and we aim to accurately measure coordinate‑wise dependencies. To address these challenges, we develop a new Fuk–Nagaev–type inequality using functional dependence measures to control the tails of the Ruppert–Polyak average of SGD.
>
> **4. Is it possible to show a central limit theorem for the iterates of the stochastic approximation process?**
>
> Thanks a lot for your great question. Yes, based on the GMC property, we can provide the central limit theorem of the SGD iterates, as well as a quenched version which holds for any arbitrarily fixed initialization. Even better than the asymptotic normality, we can quantify the rate of the Gaussian approximation as follows.
>
> **Theorem (High-dimensional sequential Gaussian approximation).** *Consider $d$-dimensional SGD iterates $\boldsymbol{\beta}\_k$ for $T$ iterations. Suppose that our Assumption 2 (stochastic Lipschitz continuity) holds for some $q>2$, the dimension $d\le cT$, for some constant $c>0$. Then, on a potentially different probability space, there exist random vectors $\\{\tilde{\boldsymbol{\beta}}\_k\\}\_{k=1}^T\overset{\mathcal{D}}{=}\\{\boldsymbol{\beta}\_k\\}\_{k=1}^T$ and independent Gaussian random vectors $\\{\tilde{\mathbf{y}}\_k\\}\_{k=1}^n$ with mean zero and covariance matrix*
> \begin{equation*}
>  \Xi=\sum\_{k=-\infty}^{\infty}\mathrm{Cov}(\boldsymbol{\beta}\_0,\boldsymbol{\beta}\_k)
> \end{equation*}
> *such that*
> \begin{equation*}
> \Big(\mathbb{E}\max\_{k\le T}\Big|\frac{1}{\sqrt{k}}\sum\_{i=1}^k\big[(\tilde{\boldsymbol{\beta}}\_i -\boldsymbol{\beta}^{\diamond}) -\tilde{\mathbf{y}}\_i\big]\Big|_2^2\Big)^{1/2} \le c'M\_{2,q}\sqrt{d\log(T)}\Big(\frac d T\Big)^{\frac{q-2}{6q-4}},
> \end{equation*}
> *where $c'$ is a universal constant that only depends on $c$ and the moment index $q$, $\boldsymbol{\beta}^{\diamond}=\lim\_{k\rightarrow\infty}\mathbb{E}[\boldsymbol{\beta}\_k]$, and $M\_{2,q}$ is defined in our Assumption 2.*
>
> The key idea to prove this Gaussian approximation is to apply the recent work by Mies and Steland (2023), in particular their Theorem 3.1. Using the GMC property of the SGD iterates, one can verify the conditions to apply this result. Ideally, when the moment index $q\rightarrow\infty$, the Gaussian approximation rate approaches $O(\sqrt{\log(T)}(d^4/T)^{1/6})$. Thus, to obtain a nontrivial bound for Gaussian approximation within $T$ iterations, we can allow dimension dependence $d=o(T^{1/4-\zeta})$ for some $\zeta>0$.
>
> **5. How strong is the $\ell_s$ strong convexity assumption in relationship to $\ell_2$ the strong convexity?**
>
> Thanks so much for another great question. First, we would like to clarify that although we suggest large $s$ with $s\approx\log(d)$ to make $\ell^s$-norm approximate $\ell^{\infty}$-norm, all of our derived theory also works for small $s$, including $s=2$ as a special case. By letting $s=2$, our Assumption 1 reduces to the standard strong convexity for $\ell^2$-norms, and the moment convergence in Theorem 2 reduces to $(\mathbb{E}|\boldsymbol{\beta}\_k-\boldsymbol{\beta}^*|\_2^2)^{1/2}=O(M\_{2,q}\sqrt{\max\\{q,2\\}\alpha}+r\_{\alpha,2,q}^k|\boldsymbol{\beta}_0-\boldsymbol{\beta}\_0^{\circ}|\_2)$, where the dependency on the dimension $d$ will be reflected by $M\_{2,q}$, $\alpha$ and $L\_{2,q}$ included in the contraction speed $r\_{\alpha,2,q}$. The moment convergence for ASGD in Theorem 3 and the tail probability inequality in Theorem 4 will follow similarly.
>
> For the case $s>2$, please kindly refer to Lemma 5 in the Appendix where we illustrate the additional requirement posted by the $\ell^s$ strong convexity assumption. We would like to provide a more detailed explanation here and will add it to our revision. In fact, there is no apparent relationship between these two conditions in general cases since different norms are involved. We therefore study the linear regression as an example. Recall that for the linear regression setting in Section 8.2, we define $\lambda\_{\min}^{(s)}=\inf\_{\|\mathbf{v}\|\_s=1}\langle\mathbf{v}^{s-1},\Sigma\mathbf{v}\rangle$. When $s=2$, $\lambda\_{\min}^{(2)}=\lambda\_{\min}(\Sigma)\ge \lambda\_{\min}^{(s)} \ge\min\_i(\Sigma\_{ii}-\sum\_{j\neq i}|\Sigma\_{ij}|)$. The rightmost "Gershgorin gap" is a universal lower bound for every $s$. The lower bound is non-trivial if $\Sigma$ is sufficiently diagonally dominant.

---

> > ### Comment · Reviewer_sNN4 · 2025-08-03
> >
> > Thank you for your rebuttal. I believe the contribution of this work is good and the extension of the existing results to prove a central limit theorem alongside this associated discussion concerning geometric moment contraction would be valuable in the camera-ready version.

---

> > > ### Author Response · Authors · 2025-08-03
> > > **Thanks a lot for your feedback!**
> > >
> > > We really appreciate your suggestions! If we are fortunate to get accepted, we will follow your advice to add the central limit theorem as well as the discussion on the geometric moment contraction to our camera-ready version.

---

### Official Review · Reviewer_1YYQ · 2025-06-28

**Clarity:** 3
**Significance:** 3
**Originality:** 3
**Rating:** 5
**Confidence:** 4

**Summary:**

This manuscript investigates the convergence of SGD and its averaged variant with constant learning-rates in high-dimensional regimes.

**Questions:**

The questions that I am concerned about have been listed in the 'Weaknesses'.

**Ethical Concerns:**

["NO or VERY MINOR ethics concerns only"]

**Final Justification:**

I believe the author(s) have thoroughly addressed my concerns. So, i should raise my score.

**Limitations:**

Yes

**Paper Formatting Concerns:**

The presentation of this manuscript satisfied the formatting requirements of NeurIPS 2025.

**Quality:**

3

**Strengths And Weaknesses:**

Strengths:

1. The authors propose two assumptions on the objective function and the stochastic gradients, which make it possible to prove the asymptotic stability in distribution in SGD iterates with any initialization. GMC is the highlights of this study.
2. The contraction of SGD and its averaged variant in max-norm and moment bounds are also derived.
3. For the averaged SGD, the corresponding tail probability inequality (in max-norm) is proved, which implies a sharp high-probability upper bound.


Weaknesses:

some typos
1. Since the abbreviation 'GMC' will be used in line 138, it should be mentioned in line 45.
2. In line 671-672 of supplementary, punctuation mark is missing in the the inequality.

major concerns
1. In a series of work by Paquette et al (see some selected papers below), they study the high-dimensional SGD for the least–squares problem. The authors also consider linear regression as an example to illustrate their theoretical results. Could you please provide a detailed comparison between these literatures and your works? I believe this would help us better understand the contributions of your research.

    [1] Courtney Paquette, Kiwon Lee, Fabian Pedregosa, Elliot Paquette. SGD in the Large: Average-case Analysis, Asymptotics, and Stepsize Criticality. 34th Annual Conference on Learning Theory (COLT), 2021.

    [2] Courtney Paquette, Elliot Paquette, Ben Adlam, Jeffrey Pennington. Implicit Regularization or Implicit Conditioning? Exact Risk Trajectories of SGD in High Dimensions. Advances in Neural Information Processing Systems 35 (NeurIPS), 2022

2. Although the authors mentioned the limitation of 'strong convexity' in Assumption 1, I am curious about whether this assumption can be relaxed to obtain the main results of this manuscript.
3. I would be glad to see more validation of the theoretical result. The authors should provide some numerical experiments to illustrate their theoretical results.


In summary, I am currently leaning towards suggesting a Borderline Accept. I look forward to receiving further clarification, which leads to a reconsideration of my evaluation.

---

> ### Author Rebuttal · Authors · 2025-07-30
>
> Thank you so much for your thoughtful evaluation and for recognizing the contributions of our work. Your feedback has helped us clarify the significance of our results. In the revision, we will fix the two typos you kindly pointed out. Please see our point-to-point responses to your major concerns below.
>
> **1. In a series of work by Paquette et al (see some selected papers below), they study the high-dimensional SGD for the least–squares problem. The authors also consider linear regression as an example to illustrate their theoretical results. Could you please provide a detailed comparison between these literatures and your works? I believe this would help us better understand the contributions of your research.**
>
> **[1] Courtney Paquette, Kiwon Lee, Fabian Pedregosa, Elliot Paquette. SGD in the Large: Average-case Analysis, Asymptotics, and Stepsize Criticality. 34th Annual Conference on Learning Theory (COLT), 2021.**
>
> **[2] Courtney Paquette, Elliot Paquette, Ben Adlam, Jeffrey Pennington. Implicit Regularization or Implicit Conditioning? Exact Risk Trajectories of SGD in High Dimensions. Advances in Neural Information Processing Systems 35 (NeurIPS), 2022.**
>
> Thank you very much for pointing us to these two interesting papers. We have carefully read them. The key techniques therein are very novel, but we believe they rely on the quadratic structure of the least-square loss, which cannot be easily extended to nonlinear cases. Our proposed theory serves as a complement to these studies. Below we provide a detailed comparison.
>
> (i) **Settings and techniques.** Paquette et al. (2021) study the least-square problem
>
> \begin{equation*}
>         \min_{x\in\mathbb{R}^d}\frac{1}{2n}\|\mathbf{a}_i\mathbf{x}-b_i\|^2, \quad \text{with }\mathbf{b}=\mathbf{A}\mathbf{\tilde x}+\sqrt{n}\mathbf{\eta},
> \end{equation*}
>
> where $\mathbf{A}\in\mathbb{R}^{n\times d}$ is a random data matrix whose $i$-th row is $\mathbf{a}_i\in\mathbb{R}^d$, $\mathbf{\tilde x}\in\mathbb{R}^d$ is a signal vector, and $\mathbf{\eta}$ contains the noise. The Hessian $\mathbf{H}=\tfrac 1n \mathbf{A}^{\top}\mathbf{A}$ is a fixed matrix (independent of the parameter $\mathbf{x}$), admitting an orthonormal eigen‐decomposition. This allows one to rewrite the SGD iteration with learning rate $\gamma$,
> \begin{equation*}
>         \mathbf{x}\_{t+1} = \mathbf{x}\_{t} - \gamma \mathbf{H} \mathbf{x}\_{t} + \text{noise},
> \end{equation*}
> into $d$ recursions individually, that is,
> \begin{equation*}
>         y\_{t+1}^{(i)} = (1-\gamma \lambda_i)y\_{t}^{(i)} + \text{noise in the i-th coordinate}, \quad i=1,\ldots,d,
> \end{equation*}
> where $\lambda_i$ is the $i$-th largest eigenvalue of $\mathbf{H}$ and $y^{(i)} = \mathbf{u}_i^{\top}\mathbf{x}$ with $\mathbf{u}_i$ the eigenvector associated with $\lambda_i$. This form evolves each coordinate $y^{(i)}$ independently as a one–dimensional linear recursion. Then, one can aggregate over $i$ to obtain
> \begin{equation*}
>         \mathbb{E}|\mathbf{x}\_{t+1}|\_2^2 = \sum\_{i=1}^d\mathbb{E}(y\_{t+1}^{(i)})^2 = \sum\_{i=1}^d(1-\gamma\lambda_i)^2\mathbb{E}(y\_{t}^{(i)})^2 + \sum\_{i=1}^d\text{noise variance in the i-th coordinate}.
> \end{equation*}
> As $d$ grows to infinity, this discrete‐time evolution can be rewritten as a Volterra integral equation (see Eq. (11) therein), which is the key technique for their theoretical results with clear illustration in Section 2. The paper Paquette et al. (2022) follows a similar strategy by adopting the Volterra integral equation (see Theorem 2 therein).
>
> Nevertheless, a general strongly-convex nonlinear loss function $f(\mathbf{x})$ has non-constant curvature, meaning that the Hessian matrix $\mathbf{H}(\mathbf{x})=\nabla^2f(\mathbf{x})$ depends on the input $\mathbf{x}$. This also implies that the overall dynamic cannot be decomposed anymore into separate disentangled univariate dynamics.
>
> Instead, in our work, we view the high-dimensional SGD as a nonlinear vector autoregressive process (VAR) and introduce the geometric-moment contraction (GMC) to study the joint convergence of the SGD vector $\mathbf{x}\_{t+1}$ as a whole, which does not rely on any diagonalization of the Hessian. This relies on the powerful concept of functional dependence measure that we introduce in Section 8.6 in the Appendix, measuring the degree of the dependence of outputs on inputs in functional systems (Wu 2005). By writing the SGD recursion into an iterated function, one can apply this tool and effectively characterize its convergence even in high-dimensional settings.
>
> (ii) **Data distributions.** Furthermore, Paquette et al. (2021) requires the data matrix $\mathbf{A}$ to be orthogonally-invariant (see Assumption 1.2). The authors mention that this is a rather strong assumption with a classical example of a matrix whose entries are generated from standard Gaussians. In addition, the "quasi-random" assumption in Paquette et al. (2022) holds Gaussian linear regression, sub-Gaussian linear designs and Gaussian random features with a linear ground truth as examples. As a comparison, our theory allows more general data distributions as long as the stochastic gradients have finite $q$-th moment for some $q\ge2$ (see our Assumption 2, Lemmas 6(iii) and 7(iii)).
>
> (iii) **Dependence on the dimension $d$.** It is a bit difficult to directly make a comparison on the dependence on $d$ since we consider different settings from these two studies. From our understanding, although Paquette et al. (2022) has relatively strong conditions $d^{\epsilon}\le n\le d^{1/\epsilon}$ for some $\epsilon\in(0,1]$ to derive the equivalence between SGD and HSGD (Theorem 1 therein), it can be relaxed in the analysis of HSGD. In our paper, the dependence on $d$ (see our Eq. (9)) is reflected by the constant learning rate $\alpha$, the moment of stochastic gradient $M\_{\log(d),q}$ and some logarithm factor $\log(d)$.
>
> In short, we would like to emphasize that our proposed theory is applicable to a wider range of nonlinear loss functions, including the least-square problem as a special case. The dependence on $d$ can be explicitly characterized for in-expectation and in-probability convergence.
>
> **2. Although the authors mentioned the limitation of `strong convexity' in Assumption 1, I am curious about whether this assumption can be relaxed to obtain the main results of this manuscript.**
>
> We greatly appreciate your comment on our limitation of strong convexity. In fact, the strong convexity in Assumption 1 cannot be relaxed for the geometric-moment contraction (GMC) result in Eq. (6). To see this, we denote the SGD recursion by an iterated random function as follows
> \begin{equation*}
> \boldsymbol{\beta}\_k=\boldsymbol{\beta}\_{k-1} - \alpha \nabla g(\boldsymbol{\beta}\_{k-1},\boldsymbol{\xi}\_k)=:f\_{\boldsymbol{\xi}\_k}(\boldsymbol{\beta}\_{k-1}).
> \end{equation*}
> We take $s=2$ and $q=2$ as a special example. Assume there exists a pair $\boldsymbol{\beta}\neq\boldsymbol{\beta}'$ such that the quantity
> \begin{equation*}
> K:=\frac{\langle \boldsymbol{\beta} - \boldsymbol{\beta}', \nabla G(\boldsymbol{\beta}) - \nabla G(\boldsymbol{\beta}')\rangle}{|\boldsymbol{\beta}-\boldsymbol{\beta}'|\_2^2}\le0.
> \end{equation*}
> For this particular pair $\boldsymbol{\beta}\neq\boldsymbol{\beta}'$, we have
> \begin{equation*}
> \frac{\mathbb{E}|f\_{\boldsymbol{\xi}}(\boldsymbol{\beta}) - f\_{\boldsymbol{\xi}}(\boldsymbol{\beta}')|\_2^2}{|\boldsymbol{\beta}-\boldsymbol{\beta}'|\_2^2} = 1-2\alpha K + \alpha^2\frac{\mathbb{E}|\nabla g(\boldsymbol{\beta},\boldsymbol{\xi}) - \nabla g(\boldsymbol{\beta}',\boldsymbol{\xi})|\_2^2}{|\boldsymbol{\beta}-\boldsymbol{\beta}'|\_2^2},
> \end{equation*}
> which is strictly larger than 1 for all $\alpha>0$, violating the GMC in Eq. (6).
>
> This GMC result is the key property such that the SGD iterates converge to a unique stationary distribution as the iteration number $k$ grows regardless of the initialization $\boldsymbol{\beta}\_0$. For the more general settings such as the non-convex loss objectives with multiple local minimums, the convergence of the algorithm will be more sensitive to the choice of initialization. For example, bad initializations may trap the iterates in some local balls. In such cases, it is unlikely to achieve the convergence to a unique stationary distribution for all initial points. However, we would like to emphasize that our proposed idea to view the SGD updates as nonlinear time series can still be useful to address the non-stationarity and complex dependency structures. Developing a notion of local GMC property will be addressed in subsequent work.
>
>
> **3. I would be glad to see more validation of the theoretical result. The authors should provide some numerical experiments to illustrate their theoretical results.**
>
> Thanks a lot for this suggestion. We fully agree. Since figures are not allowed in the rebuttal, we plan to add the results to the revision. Here, we would like to provide a detailed plan for the experiment. We plan to validate the moment convergence rate of SGD provided in Theorem 2, where we show that the SGD iterates $\boldsymbol{\beta}\_k$ forget the initialization $\boldsymbol{\beta}\_0$ exponentially fast and will converge at the rate $O(M\_{s,q}\sqrt{\max\\{q,s\\}\alpha})$. Specifically, we will generate i.i.d.\ random samples $(y\_k,\mathbf{x}\_k)$ from a linear regression model $y\_k=\mathbf{x}\_k^{\top}\boldsymbol{\beta}^* + \epsilon\_k$, where $\epsilon\_k$ are i.i.d.\ standard Gaussian random variables and $\mathbf{x}\_k$ are multivariate standard Gaussian vectors. All our technical assumptions will be satisfied in this setting. We take $s=2$ and $q=2,$ different dimensions $d$ and different learning rates $\alpha.$ In view of Lemma 7, we expect to see that the MSE scales with $\sqrt{d\alpha}$. We will also include the cases for the convergence in $\ell^{\infty}$-norm and $\ell^{s}$-norms with $s>2$ for completeness.

---

> > ### Comment · Reviewer_1YYQ · 2025-08-02
> > **Thank you for your explanations.**
> >
> > Dear author(s),
> > I have carefully read them. I believe the author(s) have thoroughly addressed my review and will raise my score. Hopefully it gets accepted.

---

> > > ### Author Response · Authors · 2025-08-02
> > > **Thank you so much for your feedback!**
> > >
> > > Dear reviewer, we would like express our sincere gratitudes to your feedback! Thanks so much for your time and support which has helped us a lot to contribute to the community!

---

### Note · Authors · 2025-08-12

We sincerely thank the reviewers for their thorough evaluations and constructive feedback. Reviewers found the paper to be clearly written, technically sound, and well-structured, with self-contained proofs and an extensive appendix that supports the main results. They appreciated the novelty and usefulness of the geometric-moment contraction framework, its successful application to both last-iterate and averaged-iterate SGD convergence in high-dimensional regimes, and its ability to deliver sharp high-probability bounds, especially for the $\ell^{\infty}$-norm. The theoretical contributions were recognized as rigorous and of broad relevance, combining convergence guarantees, moment bounds, and sharp concentration inequalities into a unified and accessible framework that can be applied to a broad range of stochastic approximation settings.

Reviewers raised valuable questions regarding presentation, assumptions, and additional clarifications. Major concerns included: **(1)** providing a more detailed comparison with related literature (e.g., Paquette et al., 2021; Paquette et al., 2022); **(2)** establishing the central limit theorem and Gaussian approximation for the distribution convergence; **(3)** clarifying the role of the $\ell^s$ strong convexity condition in Assumption 1 with comparison to the standard $\ell^2$ strong convexity; **(4)** adding the coercivity assumption to ensure the existence of the global minimum and fixing Lemma 4; and **(5)** discussing the connection between geometric-moment contraction and stationary distribution. Through our rebuttal and discussion with reviewers, we have resolved each point with detailed theoretical explanations, added comparisons with relevant works, clarified technical assumptions, and outlined planned additional experiments and examples.

For the camera-ready version, we will revise our manuscript as we outlined above. Finally, we would like to express our most sincere gratitude to all reviewers, the AC, and the SAC again for their time, expertise, and thoughtful engagement with our work. The detailed feedback not only helped us address potential ambiguities but also inspired meaningful improvements that we believe will significantly enhance the quality and impact of the final paper.

---

### Decision · Program_Chairs · 2025-09-17

**Decision:**

Accept (poster)

**Comment:**

This paper provides rigorous statistical guarantees for constant learning-rate SGD and its averaged variant in high-dimensional regimes by introducing a novel geometric-moment contraction (GMC) framework. Reviewers lauded the paper's technical strength, particularly the new GMC technique adapted from high-dimensional time series and the derivation of a sharp inequality for high-probability bounds. Noted weaknesses include an insufficient comparison to related literature, which in my view still persists after rebuttal, and questions about the strong convexity assumption, including a flawed proof for the existence of a global minimum that was resolved during discussion. Altogether, the novel techniques and strong theoretical contributions are a constructive addition to the field. Therefore, this paper is recommended for acceptance.